# Complexity and Specificity of Sec61-Channelopathies: Human Diseases Affecting Gating of the Sec61 Complex

**DOI:** 10.3390/cells10051036

**Published:** 2021-04-27

**Authors:** Mark Sicking, Sven Lang, Florian Bochen, Andreas Roos, Joost P. H. Drenth, Muhammad Zakaria, Richard Zimmermann, Maximilian Linxweiler

**Affiliations:** 1Department of Medical Biochemistry & Molecular Biology, Saarland University, D-66421 Homburg, Germany; mark.sicking@uni-saarland.de; 2Department of Otorhinolaryngology, Head and Neck Surgery, Saarland University Medical Center, D-66421 Homburg, Germany; Florian.Bochen@uks.eu (F.B.); Maximilian.Linxweiler@uks.eu (M.L.); 3Department of Neuropediatrics, Essen University Hospital, D-45147 Essen, Germany; roos@andreas-roos.de; 4Department of Molecular Gastroenterology and Hepatology, Radboud University Medical Center, 6525 GA Nijmegen, The Netherlands; Joost.Drenth@radboudumc.nl; 5Department of Genetics, Hazara University, Mansehra 21300, Pakistan; zakariaswati@gmail.com

**Keywords:** BiP, common variable immunodeficiency, congenital disorder of glycosylation, endoplasmic reticulum, neutropenia, polycystic liver disease, Sec61-channelopathies, Sec62, Sec63, SSR/TRAP complex

## Abstract

The rough endoplasmic reticulum (ER) of nucleated human cells has crucial functions in protein biogenesis, calcium (Ca^2+^) homeostasis, and signal transduction. Among the roughly one hundred components, which are involved in protein import and protein folding or assembly, two components stand out: The Sec61 complex and BiP. The Sec61 complex in the ER membrane represents the major entry point for precursor polypeptides into the membrane or lumen of the ER and provides a conduit for Ca^2+^ ions from the ER lumen to the cytosol. The second component, the Hsp70-type molecular chaperone immunoglobulin heavy chain binding protein, short BiP, plays central roles in protein folding and assembly (hence its name), protein import, cellular Ca^2+^ homeostasis, and various intracellular signal transduction pathways. For the purpose of this review, we focus on these two components, their relevant allosteric effectors and on the question of how their respective functional cycles are linked in order to reconcile the apparently contradictory features of the ER membrane, selective permeability for precursor polypeptides, and impermeability for Ca^2+^. The key issues are that the Sec61 complex exists in two conformations: An open and a closed state that are in a dynamic equilibrium with each other, and that BiP contributes to its gating in both directions in cooperation with different co-chaperones. While the open Sec61 complex forms an aqueous polypeptide-conducting- and transiently Ca^2+^-permeable channel, the closed complex is impermeable even to Ca^2+^. Therefore, we discuss the human hereditary and tumor diseases that are linked to Sec61 channel gating, termed Sec61-channelopathies, as disturbances of selective polypeptide-impermeability and/or aberrant Ca^2+^-permeability.

## 1. Introduction

The ribosome-studded or rough endoplasmic reticulum (ER) of nucleated human cells plays essential roles in protein biogenesis, calcium (Ca^2+^) homeostasis, and signal transduction (Figure 1) [1,2,3,4,5,6,7,8,9,10,11,12,13,14,15,16,17]. Approximately one hundred ER proteins are involved in two aspects of protein biogenesis at the ER, protein import and protein folding or assembly (Table 1). Additional ER proteins are involved in (i) structurally shaping ER sub-domains [18,19,20,21,22], (ii) vesicular transport for the delivery of native non-ER proteins to other organelles with a function in endo- or exocytosis [23,24,25], (iii) ER-associated protein degradation (ERAD) [26,27,28,29,30] or ER-phagy [31,32,33,34], (iv) signal transduction pathways in response to unfolded proteins (UPR) or the ER-stress induced pathway of apoptosis [35,36,37,38,39,40], and (v) channels, receptors and pumps, which facilitate the controlled exchange of Ca^2+^ between the ER and other intra- and extracellular compartments [4,7,8,9,14,15,16,17,41,42,43,44]. Proteins that need to be named in these respects are, for example, atlastins and reticulons for ER morphology (i) [18,19,20,21,22], SNARE proteins, and small GTPases for vesicular transport (ii) [25], Hrd1—and possibly Sec61 complex—together with the ATPase valosin-containing protein or VASP for ERAD [26,27,28,29,30,45] and FAM134B -and possibly Sec62- for ER-phagy (iii) [31,32]. IRE1, ATF6 and PERK for UPR and cytosolic CHOP are relevant for apoptosis (iv) [35,36,37,38,39,40], and inositol-1,4,5-triphosphate receptor (IP3R), sarcoplasmic/endoplasmic reticulum ATPase (SERCA), STIM plus ORAI and Sigma-1-receptor for Ca^2+^ homeostasis (v) [4,5,6,7,8,9,10,11,12,13,14,15,16,17,46]. In relation to Ca^2+^ it is noteworthy that there is also un-controlled or passive Ca^2+^-efflux from the ER, which has also been termed Ca^2+^-leakage, was recently linked to ATP/ADP exchange across the ER membrane and may contribute to apoptosis when a cell is sacrificed in order to save the organism [4,5,6,7,8,9,10,11,41,42,43,44]. Some of the proteins named here will come up below or will be discussed in further detail in other articles of this Special Issue. 

The heterotrimeric Sec61 complex of the ER membrane represents the major entry point for precursor polypeptides into the membrane or lumen of the ER (Figure 1 and Figure 2) [47,48,49,50,51,52,53,54,55,56,57,58,59]. Therefore, it can form an aqueous polypeptide-conducting channel, which also allows the passage of Ca^2+^ in the opposite direction [60,61,62,63,64,65,66,67,68,69]. The channel exists in two conformations, an open and a closed state that are in a dynamic equilibrium with each other (Figure 2). The closed conformation is impermeable even to Ca^2+^. Thus, the Sec61 complex is a precursor-gated channel, which operates either coupled to translation (in co-translational transport) or after completion of translation (in post-translational transport). Sec61 gating to the open state is not solely facilitated by its substrates, the amino-terminal signal peptides (SPs) or transmembrane helices (TMHs) of precursor polypeptides [70,71,72,73,74,75], it is also supported by ribosomes in co-translational transport [52,55] and/or by several Sec61 interaction partners, such as translocon-associated protein or TRAP complex [53,56,76,77] and Sec62/Sec63 complex in cooperation with BiP [69,78,79,80,81], respectively. Here, the latter are defined as allosteric effectors of the channel since they interact with the complex at sites, which are distinct from the precursor binding sites. Channel closing also appears to be facilitated by allosteric effectors, such as the ER-lumenal BiP [69] and/or the cytosolic Ca^2+^-calmodulin (CaM) [68]. In our opinion, Sec61 channel gating can best be described in analogy to an enzyme-catalyzed reaction (Figure 3): Channel opening and closing represent two energetically un-favorable reversible reactions and the substrates and effectors are the catalysts, which lower the activation energy for the required conformational transitions by binding to the Sec61 complex [57,58]. 

The Hsp70-type molecular chaperone immunoglobulin heavy chain binding protein or BiP [82] does not only support Sec61 channel opening for ER protein import [69,78], but also can bind to incoming precursor polypeptides and act on these as a molecular ratchet [83]. Thus, typical for an Hsp70, the ATP- and Ca^2+^-dependent BiP modulates the conformation of a folded protein complex, the Sec61 channel, plus interacts with a more or less unfolded polypeptide chain as it emerges from the Sec61 channel, thereby contributing to a unidirectional or irreversible transport process. Also typical for an Hsp70, both BiP activities involve an ATPase cycle and their own allosteric effectors, i.e., J-domain-proteins (JDPs) [84] or Hsp40-type co-chaperones, termed ERj- or ERdj-proteins, and nucleotide exchange factors (NEFs). Following the same principles and interactions, BiP also plays a central role in folding and assembly of newly-imported polypeptides, such as heavy and light chains of immunoglobulins in the plasma cells of the immune system [85], and supports efficient Sec61 channel closing to preserve Ca^2+^ homeostasis [69]. In addition, BiP is a key player in various Ca^2+^-dependent and -independent signal transduction pathways, which report on ER energy- and protein-homeostasis (proteostasis), as reviewed in other articles of this Special Issue.

In this article, we zoom in on the question of how the functional cycles of BiP and the Sec61 channel are intertwined and which allosteric effectors of the two are involved in these reactions. Furthermore, we discuss the human hereditary and tumor diseases as well as human pathogens that are linked to Sec61 channel gating, the Sec61-channelopathies, as disturbances of selective polypeptide-impermeability and/or Ca^2+^ permeability of the ER membrane and highlight the importance of the functionality of the system [86].

## 2. The Human Sec61 Translocon

Protein import into the ER is the first step in the biogenesis of precursors of about 10,000 different soluble and membrane proteins of nucleated human cells, which amounts to about 30% of the proteome [1,2,3]. All these proteins fulfill their functions either in the membrane or lumen of the ER (plus the connected nuclear envelope), in one of the organelles of the pathways for endo- and exocytosis (i.e., ERGIC, Golgi apparatus, endosome, lysosome), in lipid droplets or at the cell surface as plasma membrane- or secretory-proteins. ER protein import involves the two stages of membrane targeting and insertion of nascent membrane proteins into or translocation of soluble precursor polypeptides across the ER membrane. Typically, both processes depend on SPs or TMHs at the amino-termini of the precursor polypeptides [70,71,72,73,74,75]. In general, these SPs have a tripartite structure. They comprise a more or less positively charged amino-terminal or N-region, a central hydrophobic or H-region and a slightly polar carboxy-terminal or C-region (Figure 4). Other than that, they do not have sequence homologies and, as a matter of fact, show quite some variability with respect to length (15–50 amino acid residues) as well as overall properties (see below). Interestingly, various human hereditary diseases are the result of single point mutations in the SPs of certain precursor polypeptides (such as preproinsulin and preprorenin), which result in failure of these SPs to deliver their otherwise functional mature forms to the correct cellular location, thus, highlighting the fact that these amino-terminal SPs were fine-tuned to their respective receptors by evolution [87,88,89]. In addition, insertion of SPs may occur co- or post-translationally and are facilitated by various pathways and components, which reside in the cytosol and the ER membrane or lumen, respectively (Table 1). 

### 2.1. Entry of Precursor Polypeptides into the ER

The heterotrimeric Sec61 complex of the ER membrane represents the entry point for most precursor polypeptides with a SP or TMH into the membrane or lumen of the ER (Figure 1, Figure 2 and Figure 3) [48,49,50]. Cryo-electron tomography (CET) of cells or isolated ER-derived vesicles (rough microsomes) depicts the Sec61 as a large multicomponent assembly in association with translating ribosomes and the membrane-embedded TRAP (also termed SSR) complex and oligosaccharyltransferase (OST), the enzyme complex that catalyzes N-linked glycosylation (Figure 1c) [51,53,54,56]. This super-complex was termed Sec61 translocon and can insert into the membrane or fully import into the lumen an amazing variety of precursor polypeptides (Figure 4). These precursors mature to (i) membrane proteins with one, two or multiple TMHs and with their amino-termini either in the cytosol or the ER lumen, (ii) GPI-anchored membrane proteins or (iii) soluble proteins in the ER lumen, such as secretory proteins. Membrane insertion is either mediated by a cleavable amino-terminal SP or the amino-terminal TMH of the nascent precursor polypeptide. The import of soluble polypeptides into the lumen is invariably mediated by cleavable amino-terminal SPs. GPI-anchored membrane proteins are imported in analogy to soluble polypeptides and, concomitantly, modified by another multimeric enzyme complex called GPI transamidase. In this case, membrane integration is limited to the lipid moiety of the GPI-anchor [90]. Cleavable SPs are removed from the inserting or incoming precursor polypeptides by yet another heteromultimeric enzyme, the signal peptidase complex (SPC) [91,92].

### 2.2. Targeting of Precursor Polypeptides to the ER

Prior to ER entry, however, precursor polypeptides have to be targeted to the ER membrane [93]. In case of the Sec61-dependent ER import, co-translational ER targeting is mediated by the cytosolic ribonucleoparticle signal recognition particle (SRP) and its heterodimeric receptor in the ER membrane, termed SRP receptor or SR (Table 1) [94,95,96,97,98]. Another binary targeting system consisting of a single ribosome-associating component (SND1) and a heterodimeric membrane receptor (SND2 plus SND3) directing precursor polypeptides to the Sec61 complex was identified in yeast and named SRP-independent (SND) pathway [99,100,101,102,103,104]. In human cells only SND2 was found to be conserved compared to yeast and the other components still await identification. In human cells this targeting pathway can for example be used by small presecretory proteins (i.e., precursors with less than 100 amino acid residues, such as preproapelin and prestatherin) [102,103]. Thus, for posttranslational import of small precursor proteins via the Sec61 complex, ER targeting can occur via the SND pathway or via direct contact with the Sec61 complex and its associated components (Sec62). In addition to the above-mentioned membrane proteins, the ER membrane also contains hairpin- and tail-anchored or TA-membrane proteins, which depend on dedicated components and post-translational pathways for their membrane insertion (Figure 4). The TRC-pathway (GET-pathway in yeast) handles TA proteins and the PEX3-dependent pathway at least one hairpin protein, which is destined to lipid droplets [105,106,107,108,109,110,111,112,113,114,115,116]. In case of the TRC- and PEX-pathways, targeting to these membrane components is mediated by the Bag6 complex plus additional cytosolic factors and PEX19, respectively. For reasons of clarity, this article will more or less ignore the latter two pathways and, instead, focus on Sec61-dependent import. 

Notably, however, one general lesson from the analysis of these pathways is that they are not strictly separated from each other and that there are at least some precursor polypeptides, which can be handled by more than one pathway. Some small human presecretory proteins (such as preproapelin), for example, can be targeted to the Sec61 complex via the SRP-, SND-, and TRC-pathway or directly via Sec62 [102]. Furthermore, at least some TA-membrane proteins (such as Sec61β and RAMP4) can be targeted to the membrane via the same three pathways as small presecretory proteins [100]. Thus, there is redundancy in these three targeting systems and they can substitute for each other as a backup at least to a certain extent. Another general lesson is that not all amino-terminal SPs and TMHs, which are involved in ER targeting and import of precursor polypeptides, were created equal, i.e., some have special requirements, which is not surprising considering the large variety of precursor polypeptides (Figure 4). This is where allosteric effectors of the Sec61 complex (BiP together with Sec62/Sec63 complex or TRAP complex) (Figure 2 and Figure 5) and auxiliary membrane protein insertases (EMC and TMCO1 complex) join the game [117,118,119,120,121,122,123,124,125,126]. In order to insert into the membrane or import such a large variety of different precursor polypeptides, the Sec61 complex can form a relatively promiscuous and wide aqueous polypeptide-conducting channel, which is supported by its overall structural design and described next.

### 2.3. Structure of the Sec61 Complex

The structure of the human Sec61 complex was first deduced from the X-ray crystallographic analysis of the ortholog archaean SecY complex by T. Rapoport and colleagues (Figure 1d) [127]. The high sequence conservation of the SecY and Sec61 subunits indicated that their architecture and dynamics are evolutionarily conserved, which was since confirmed by various subsequent cryo-electron microscopy—(cryo-EM)— studies on detergent-solubilized or reconstituted ribosome-bound SecY or Sec61 complexes. Accordingly, the central polypeptide-conducting channel forming subunit (Sec61α − or correctly Sec61α 1 because there also is an uncharacterized Sec61α 2 coded by the human genome) comprises ten TMHs, which are connected by four cytosolic plus five ER lumenal loops (Figure 5). The complex is arranged in pseudo-symmetrical amino- and carboxy-terminal halves with an overall hourglass-shaped structure and a central constriction. This constriction is called pore ring and is sealed by a ring of the bulky hydrophobic residues I81, V85, I179, I183, I292 plus L449 in TMHs 2, 5, 7, and 10. A short, flexible helix between residues F62 and S82 of the luminal loop 1 was termed plug helix. The two halves are connected by a “hinge” region and subunits Sec61β and Sec61γ are located on the outskirts of the Sec61 complex and comprise one tail anchor each. Strikingly, two distinct conformational states of the Sec61 channel could be distinguished, which differ in the relative positioning of the amino- and carboxy-terminal halves of Sec61α (Figure 2). These states either allow or prevent lateral access of amino-terminal SPs or TMHs of precursor polypeptides from the central pore into the lipid bilayer through the so-called lateral gate, which is formed by TMHs 2 and 7 of Sec61α (Figure 1d and Figure 2). Furthermore, they either do or do not connect the cytosol and ER lumen via an aqueous channel formed by Sec61α.

Subsequent structural determination of programmed ribosome-Sec61 complexes suggested a series of events upon arrival of a nascent precursor polypeptides [52,55]. Accordingly, in co-translational transport, the closed Sec61 complex is primed by binding of the ribosome to cytosolic loops 6 and 8 of Sec61α as well as the amino-terminus of Sec61γ, unveiling a hydrophobic patch in the cytosolic funnel formed by Sec61α, which involves the residues V85, L89, I179 plus I293 from TMHs 2, 5 and 7, 5. This patch in vicinity to the lateral gate serves as an interaction site for an incoming SP, more precisely, the H-region within the SP of the precursor polypeptide. This hydrophobic interaction supports the rigid body movement of the amino- and carboxy-terminal halves of Sec61α and the channel becomes fully open with the pore ring widened and the plug displaced. The open Sec61 channel than allows the precursor polypeptide axial access to the ER lumen or lateral access into the membrane. The cryo-EM data as well as biochemical analyses also demonstrated that even in co-translational translocation, a considerable stretch of certain nascent precursor polypeptides can accumulate at the interface between the ribosome and the Sec61 channel without necessarily aborting translocation [128]. This indicated that nascent precursor polypeptide chain elongation does not or at least not always provide the driving force for translocation.

CET of Sec61 translocons in rough microsomes derived from human cell lines and even in intact cells has given further insight into the architecture and dynamics of the Sec61 channel in its physiological setting (Figure 1c) [53,54,56,129]. The atomic model of the solubilized ribosome-bound Sec61 complex [52], opened laterally by SPs, was easily docked into the CET density, defining the position and conformation of Sec61α in the center of the native translocon. Furthermore, weak helical density in front of the lateral gate in the CET density map confirmed the positioning of SPs observed after detergent solubilization of ribosome-Sec61 complexes. The Sec61 channel was found in a laterally open conformation, possibly implying that the Sec61 channel remains laterally open throughout the complete process of protein translocation. However, at this point the aqueous pore in the center of the channel can be expected to be occupied by the polypeptide chain in transit and, therefore, impermeable to ions.

**Figure 5 cells-10-01036-f005:**
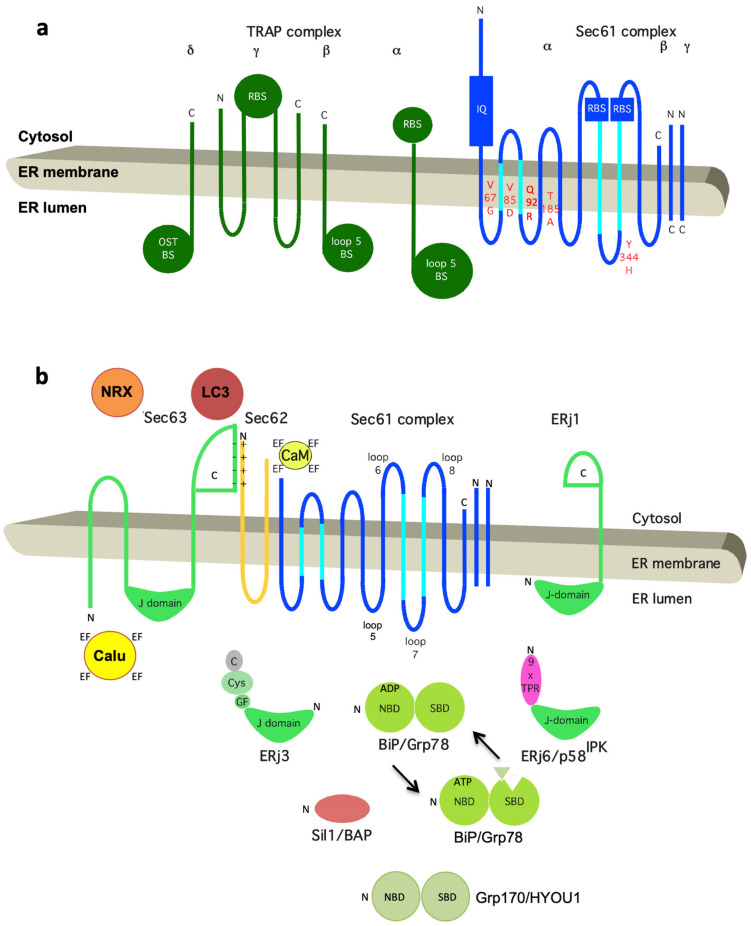
Topology and functionally relevant domains of the heterotrimeric Sec61 complex and its allosteric effectors TRAP, BiP, Sec62, and Sec63. The membrane topology of the three subunits of the mammalian Sec61 complex indicates binding sites (termed BS) for the ribosome (R, cytosolic loops 6 and 8 of Sec61α) and the TRAP complex (loop 5 of Sec61α) in (**a**) and for Ca^2+^-calmodulin (CaM, IQ) and BiP (loop 7 of Sec61α) in (**b**). The relevant allosteric effectors of BiP (ERjs and NEFs) are also shown in (b), as are additional interaction partners of Sec62 (LC3) [32] and Sec63 (nucleoredoxin or NRX, calumenin or Calu) [130]. Recent work demonstrated that IRE1a interacts with Sec61 and connects protein translocation and Ca^2+^ leakage with UPR. Furthermore, functional domains (J-domain, NBD, SBD, TPR) plus motifs (IQ, EF hand) and disease-associated mutations of Sec61α are indicated (in red; amino acid residues are given in single letter code). C, carboxy-terminus; N, amino-terminus. Notably, recent 3D reconstructions after single particle cryo-electron microscopic analysis of the yeast SEC complex showed that in the post-translationally acting Sec61 complex, the Sec62/Sec63 sub-complex interacts with the cytosolic loops 6 and 8 on the cytosolic face of the Sec61 complex, and that the ER luminal domain of Sec63 interacts with ER luminal loop 5 of Sec61α.

### 2.4. Dynamics of the Sec61 Complex

Originally, the dynamics of the Sec61 channel were observed in single-channel recordings from planar lipid bilayers, which were derived from canine pancreatic rough microsomes, after artificial release of nascent precusor polypeptides from membrane bound ribosomes by puromycin. This technique was introduced by S. Simon and G. Blobel in 1989 to the field [47], i.e., at a time where the Sec61 complex had not yet been discovered, and suggested a pore of about 10 Angstrom. Almost fifteen years later and after the discovery of the complex by T. Rapoport and R. Schekman, respectively, this approach was reproduced and adapted to purified and reconstituted canine pancreatic Sec61 complexes [61]. The complex was characterized as a highly dynamic aqueous pore with pore diameters ranging from 12 to 22 Angstrom that is (i) initially opened by SPs of defined ribosome-nascent precursor polypeptide chain-complexes (so-called RNCs) or fully-synthesized small presecretory proteins, (ii) subsequently occupied and sealed by the polypeptide chain in transit, (iii) transiently open upon release of the chain in transit and (iv) then closed, which allows a new translocation cycle to begin. The open channel was found to be permeable to various ions, including Ca^2+^, and even small molecules [67,68,69,131,132,133]. 

A priori and supported by these dynamic studies, the static structural analysis of Sec61 complexes in different states and many in vitro studies on ER protein import, it is clear that the opening of the precursor polypeptide-conducting Sec61 channel during early steps of ER protein import is mediated by SPs and TMHs [134]. Typically, the latter first approach the cytosolic funnel of the Sec61 channel [57,70,93,135]. Next, they start sampling the cytosolic funnel of the Sec61 channel as brilliantly simulated and visualized by Zhang and Miller [136] for co-translational transport. According to these simulations, sampling in the Sec61 channel pore is affected by deleterious charges, hydrophobicity, mature protein length, arrest peptides, or poly-proline motifs in the precursor polypeptides and translation speed, which is dependent on pause sites, rare codons or hairpins in the mRNA. For productive SP or TMH insertion into the Sec61 channel and concomitant complete opening of the Sec61 channel, a high hydrophobicity, i.e., low ΔG^pred^ value for the H-region were found to be conducive [135,136,137]. H-region hydrophobicity of the SP or TMH is recognized by the hydrophobic patch in the Sec61α TMHs 2 and 7, which line the lateral gate of the channel [127]. Typically, the SP- and TMH-orientation in the Sec61 channel follows the positive inside rule [138,139,140], i.e., positively charged residues in the N-region support loop insertion (N_cytosol_-C_ER-lumen_), while positively charged side chains downstream of the SP or TMH interfere with loop insertion and favour head-on insertion (N_ER-lumen_-C_cytosol_), which can be followed by a so-called “flip turn” (Figure 4) [135,141]. For SPs with low overall hydrophobicity in combination with high glycine- plus proline-content full Sec61 channel opening in co-translational transport is supported by the TRAP-complex [142]. To accommodate SPs with low H-region hydrophobicity in combination with detrimental features within the mature part, full Sec61 channel opening is supported by the Sec62/Sec63-complex with or without BiP involvement in co- and post-translational transport [102,143]. Notably, lower SP hydrophobicity has also been observed to be decisive for Sec62p/Sec63p-involvement in post-translational ER protein import in yeast [144]. 

### 2.5. Auxiliary Factors of the Sec61 Complex

The first hints on participation of additional components in co-translational protein transport came from the analysis of ribosome-associated ER membrane proteins present in detergent extracts of canine pancreatic rough microsomes. The term ribosome-associated membrane proteins (RAMPs) was coined for this class of membrane proteins after their solubilization in the presence of high salt concentrations [48,49]. By definition, the Sec61 complex is a RAMP, and so are RAMP4, TRAP, and OST (Table 1). More recently, ERj1 and Sec62 were characterized as RAMPs, although their ribosome association is seen only under more physiological salt concentrations and therefore may be more dynamic compared with the high-salt resistant RAMPs [145,146,147,148]. More information on the composition of the native protein transport machinery in the ER membrane came from fluorescence resonance energy transfer experiments, which employed fluorescently labeled antibodies against transport components, permeabilized MDCK cells, and fluorescence microscopy. According to this cell biological strategy, Sec61α1, Sec61β, Sec62, and ERj1 are RAMPs [147,148,149]. Furthermore, this approach demonstrated that SR, the TRAP complex, and translocating chain-associating membrane (TRAM) protein are permanent nearest neigbours of Sec61 complexes. Additional cross-linking data suggested that SR and Sec62 interact with Sec61α in a mutually exclusive manner and may use the same binding site at the cytosolic amino-terminus [150]. 

#### 2.5.1. Allosteric Effectors of the Sec61 Channel for Channel Opening

It is clear that some amino-terminal SPs or TMHs are strong enough to trigger immediate Sec61 channel opening on their own (such as the SP of bovine preprolactin), particularly after the ribosome has already primed the channel. However, precursor polypeptides with weak SPs involve auxiliary components in Sec61 channel opening and, therefore, in facilitating insertion of precursor polypeptides into the Sec61 complex, such as the ER-lumenal chaperone BiP together with the Sec62/Sec63 complex or the TRAP complex (Table 1) [142,143]. Alternatively, the auxiliary components may support the above-mentioned flip turn of the SP in case of an original erroneous head-first insertion. This view is based on the observations in both the yeast and the mammalian system that these auxiliary components can facilitate the flip turn of TMHs, i.e., affect the topology of TMHs that do not promote a specific initial orientation of membrane protein precursors in the membrane or to mediate topology of moderately hydrophobic signal anchor proteins, e.g., in particular type II membrane proteins that undergo the flip turn for reversing the initial type I orientation [138,139,141,144,151,152]. Based on in vitro and in cellulo experiments the concept emerged that TRAP and Sec63 plus BiP facilitate Sec61 channel opening in a substrate specific manner [76,77,78,79,80,81]. By definition, precursor polypeptides with weak SPs or TMHs were found to be affected by depletion of either component (Figure 5) [69,102,153,154,155,156,157]. Based on only a small set of model precursor polypeptides (such as preproapelin, pre-prion protein, and pre-ERj3) and import into the ER of semi-permeabilzed or intact human cells, the distinguishing factor that determines the requirement for BiP and Sec63 was suggested to be a short and rather apolar signal peptide in combination with detrimental clusters of positive charges in the mature part [102,129,156]. We suggest that the positively charged side chains downstream of the SP interfere with loop insertion of the SP and, therefore, increase the energetic barrier for Sec61 channel opening (Figure 3). The TRAP complex was observed in similar in vitro transport studies to stimulate translocation of specific proteins (such as the prion protein) indicating that there may also be redundancy at the level of the allosteric Sec61 effectors in channel opening [76,77,153,154,155,156,158,159,160,161]. More recently, the combination of siRNA mediated depletion of a certain transport component from human cells with subsequent cellular protein abundance analysis characterized SPs with comparatively longer but less hydrophobic H-regions and lower C-region polarity as Sec62/Sec63 dependent and above-average glycine-plus-proline content and below-average hydrophobicity of SPs as feature for TRAP dependence [142,143].

#### 2.5.2. Additional Auxiliary Factors of the Sec61 Complex

As noted before, several additional proteins in the mammalian ER membrane also can be considered as auxiliary components, apparently, without affecting Sec61 channel gating, most notably translocating chain-associated membrane protein or TRAM (Table 1) [162,163,164,165,166,167,168,169,170,171]. In the case of TRAM, precursors with a long N-region as well as long H-region of the SP showed a low TRAM dependence in in vitro experiments, which was since supported by the above-mentioned in cellulo experiments. Interestingly, there is a second TRAM in mammalian cells, termed TRAM2, which can invert the topology of TMHs that do not promote a specific initial orientation in the membrane [172,173], which is reminiscent of TRAP and Sec62/Sec63 (see above). We note that TRAM2 was characterized as SERCA interaction partner, i.e., also has a connection to Ca^2+^-homeostasis. In contrast to Sec61, TRAP, and OST, TRAM was not identified by CET in native ER membranes, which were derived form human cells after component depletion, probably because of its almost complete membrane embedding. Nevertheless, it was suggested to represent the density, which is consistently found opposite of the lateral gate of the Sec61 channel [51]. So far, TRAM function remains poorly defined. Based on its TLC domain, however, it was proposed to affect bilayer thickness and/or phospholipid packing in the vicinity of the lateral gate in order to support lateral exit of SPs and TMHs, in analogy to bacterial YidC and mitochondrial Oxa1 [171,174,175]. 

However, notably, WRB (Get1 in yeast), EMC3 and TMCO1 are also considered as YidC and Oxa1 homologs (Table 1) [124]. ER membrane protein complex (EMC) was first identified in yeast and later in human cells as a heteromultimeric protein complex with six and ten subunits, respectively. Biochemical and cellular characterization of the decameric EMC characterized it as both stand-alone insertase for example for TA- membrane proteins with a moderately hydrophobic transmembrane helix and as helper membrane protein insertase in synergy with the Sec61 complex for the insertion of critical TMHs of polytopic membrane proteins [117,118,119,120,121]. It was proposed to cause local membrane thinning. Besides this stable decameric protein complex TMCO1 has been shown to be in contact with the Sec61 complex and to facilitate membrane protein insertion (Table 1) [124,125]. In partial analogy with YidC, TMCO1 was originally found in association with both ribosomes and the Sec61 complex. Under these conditions, it forms a transient pentameric complex with four additional subunits and functions as either stand-alone or auxiliary membrane protein insertase to the Sec61 complex for TMHs with insufficient hydrophobicity. As stated above, several small human precursor polypeptides were observed to translocate post-translationally and ribosome-independently into the human ER. For a subset of them, ER-targeting was reported to occur independently of SRP and SR and to alternatively involve cytosolic TRC40 (Get3 in yeast) and its heterodimeric receptor in the ER-membrane (WRB/CAML) (Get1/Get2 in yeast), which can act as signal peptide recognition proteins in post-translational ER-targeting to the Sec61 complex [102]. Notably, TA-membrane proteins, such as Sec61β, may also be targeted to the membrane via the same three pathways as small presecretory proteins [100]. In fact, the TRC- pathway has its primary role in both targeting TA-membrane proteins to the ER membrane and facilitating their membrane integration [106,107,108,109,110,111,112,113]. It remains to be seen whether or not the SND-pathway also has an additional stand-alone membrane protein insertase activity.

#### 2.5.3. Structural Considerations

We note that a permanent association of ribosome-associated Sec61 complexes with TRAP and OST was confirmed in the three-dimensional (3D) reconstructions after CET of native translocons in ER membrane vesicles, derived from canine pancreas or various human cells and even intact cells (Figure 1b,c) [53,54,56,129]. Interestingly, all ribosome-associated Sec61 complexes were routinely found to be associated with TRAP, irrespective of the cellular origin of the native complexes. Mammalian TRAP is a heterotetrameric membrane protein complex, with three subunits (α, β, δ) predicted to comprise one TMH plus one lumenal domain each, while TRAPγ likely comprises a bundle of four TMHs plus a central cytosolic domain (Figure 5a) [56,161]. This ensemble of TMHs appears to be flanking both Sec61γ and the carboxy-terminal half of Sec61α, and the cytosolic domain interacts with the ribosome via ribosomal protein eL38 and a short RNA expansion segment. Significantly, the heterotrimeric ER-lumenal segment of TRAP reaches across the central Sec61 channel where the δ-subunit contacts OST and the α - and β -subunits contact ER lumenal loop 5 in the hinge region between the amino- and carboxy-terminal halves of Sec61α. In this position, the ER lumenal domain of TRAP may be able to act in a chaperone-like fashion on the conformational state of Sec61α or as a molecular ratchet on incoming precursor polypeptides into the ER lumen or both, in possible analogy to BiP (see below). We note that various algorithms predict a beta sandwich fold for the ER lumenal domains of TRAP’s α- and β- subunits and that TRAPα was also characterized as Ca^2+^-binding protein [158].

As discussed before, the Sec61 complex imports polypeptides either co-translationally or post-translationally. Structural data on the organization of the post-translationally acting mammalian Sec61 complex are currently missing. However, recent studies highlighted the architecture of the detergent solubilized, unoccupied as well as active post-translational translocon from yeast by cryo-EM [176,177,178,179]. In yeast, the fully assembled post-translational translocon represents a heptameric protein ensemble, the SEC complex. In the SEC complex the heterotrimeric Sec61 complex is associated with the heterotetrameric Sec62/63/Sec71/Sec72 complex. These data provided first insights into how the SEC complex is arranged to allow priming and gating of the Sec61 complex and support transport of post-translationally transported precursor polypeptides. Most striking was the extensive interaction between Sec63 and the Sec61 complex including contacts in their cytosolic, membrane and luminal domains. Specifically, the cytosolic Brl domain of Sec63 interacts with loops 6 and 8 of Sec61α, which form the ribosome docking site in co-translational transport. Interestingly, as predicted for the interaction of the TRAPα/β subunits with the Sec61 complex, the Brl domain of Sec63 shows a canonical beta-sandwich fold for an antigen-antibody-like binding to loop 6. In the membrane, Sec63 (TMH 3) contacts all three subunits of the Sec61 complex in the hinge region opposite to the lateral gate including TMHs 5 and 1 of Sec61α as well the membrane anchors of Sec61β and Sec61γ. In addition, the short luminal amino-terminus of Sec63 appears to intercalate on the luminal side of the channel between the Sec61α hinge loop 5 and Sec61γ. Therefore, binding of the Sec62/63 complex to the Sec61 complex was proposed to induce a fully open channel that readily accommodates even weak SPs [180]. In the substrate-occupied SEC complex the SP in transit was additionally flanked by the two Sec62 TMHs [178]. Thus, while Sec63 seems to assist opening of the Sec61 complex from a position opposite of the lateral gate, the Sec62 protein resides close to the lateral gate possibly welcoming the SP to the hydrophobic environment of the membrane.

However, this scenario unlikely reflects the complete picture in human cells, since the human Sec62/Sec63 complex was found to be involved in co-translational transport of certain precursor polypeptides (pre-prion protein and pre-ERj3) and to be strictly BiP-dependent. Briefly, in the case of pre-ERj3 deletion of the carboxy-terminal cluster of positive charges in Sec63, the Sec62 interaction site, and mutation of the HPD-motif in the J-domain of Sec63, the BiP interaction site, resulted in reduced import in HeLa cells, as had been observed for post-translational import of preproapelin (Figure 5b). In comparison to its yeast ortholog, the mammalian Sec62 protein experienced a gain of function and, therefore, is able to interact with the ribosome near the ribosomal exit tunnel and to support, in collaboration with Sec63 and BiP, the co-translational transport of the precursors of ERj3 and prion protein [143,148,156]. In addition, the BiP binding site in Sec61α was characterized as a di-tyrosine motif-containing mini-helix in ER luminal loop 7 (Figure 5b) [69,102]. Interestingly, homozygous mutation of tyrosine 344 to histidine in this loop 7 is linked to Diabetes mellitus in mice (see below) and compromises ER co- and post-translational import of Sec63- plus BiP-dependent precursor polypeptides, such as preproapelin and pre-ERj3, when introduced into HeLa cells (as does the Y343H mutation). Notably, in vitro reconstitutions demonstrated that the yeast SEC complex needs support from the Hsp70 chaperone of the ER lumen, Kar2p or BiP, for efficient post-translational transport [181,182,183,184]. This idea was subsequently dismissed on the basis of a model translocation reaction that allowed precursor movement through the yeast Sec61 complex in detergent solution. We propose that the BiP-dependent Sec61 channel gating may have been facilitated by the detergent in this artificial translocation system. 

## 3. Gating of the Sec61 Channel by BiP

The Hsp70-type molecular chaperone immunoglobulin heavy chain binding protein or BiP, which was discovered by I. Haas, also goes through a cycle of open and closed conformation [82,185,186,187,188,189,190,191,192,193,194,195,196]. However, in this case the description refers to the state of the substrate-binding domain (SBD). It also involves allosteric effectors in its conformational changes and, in contrast to Sec61, the hydrolysis of ATP. It does not only support Sec61 channel opening for ER protein import [69,78,102,197,198], but also can bind to and act on the incoming precursor polypeptide as a molecular ratchet [83]. Thus, typical for an Hsp70, the ATP- and Ca^2+^-dependent BiP modulates the conformation of a folded protein complex, the Sec61 channel, plus interacts with a more or less unfolded polypeptide chain as it emerges from the Sec61 channel, thereby vectorizing the transport process. These two kinds of substrates (folded and unfolded substrates) have previously also been observed for various other Hsp70s, such as sigma factor 32 in the bacterial cytosol and clathrin triscelions in the cytosol of human cells. Also typical for an Hsp70, both BiP activities involve an ATPase cycle, where the chaperone goes through states of substrate trapping in its ADP-bound state and substrate release in its ATP-bound state (Figure 5b). Furthermore, these activities involve JDPs, termed ERj- or ERdj-proteins, which stimulate the hydrolysis of BiP-bound ATP [84,199], and NEFs [200,201,202,203,204,205,206], which stimulate the exchange of ADP for ATP, thereby affecting BiP conformations allosterically [193,194]. Following the same principles and interactions, BiP also plays a central role in folding and assembly of newly-imported polypeptides, such as heavy and light chains of immunoglobulins in plasma cells of the immune system [85,185,186], and supports efficient Sec61 channel closing to preserve Ca^2+^ homeostasis [69,207]. In addition, BiP is a key player in various Ca^2+^-dependent and -independent signal transduction pathways, which report on ER energy homeostasis and proteostasis, respectively, and can first increase ATP/ADP exchange between ER and cytosol, next increase the folding- and ERAD-capacity via UPR and, at last, switch to apoptosis under conditions of ever increasing protein mis-folding or ER stress [35,36,37,38,39,40,41,42,43,44,208].

### 3.1. Structure and Dynamics of BiP

There are two Hsp70-type chaperones in the human ER (BiP and Grp170) but, more accurately, Grp170 is a Hsp110 protein family member [78,82,201,202,203,204]. Hsp70-type molecular chaperones, such as BiP, bind reversibly to substrate polypeptides via their carboxy-terminal substrate-binding domains (SBDs) (Figure 5b). Upon ATP hydrolysis the carboxy-terminal extension of the SDB, termed the lid domain, traps the substrate. Typically, BiP substrates are hydrophobic oligopeptides of loosely- or un-folded polypeptides. Binding of a substrate to the SBD inhibits unproductive interactions of the polypeptide and, thereby, favors productive folding and assembly, which occur concomitantly with release from BiP. In addition, BiP can also regulate the activities of folded polypeptides, i.e., induce conformational switching of a substrate (here, Sec61α). This binding and release of substrates by BiP are facilitated by interaction of its carboxy-terminal SBD and its amino-terminal nucleotide-binding domain (NBD), which are connected via the linker region. NBD-conformation and BiP’s ATPase cycle are modulated by different allosteric effectors [184,185,186,187]. The ATP-bound state of BiP has a low affinity for substrates. In contrast, the ADP-bound state has a high substrate affinity. ERjs stimulate the ATPase activity of BiP and favor substrate trapping. NEFs of the ER lumen stimulate the exchange of ADP for ATP and induce substrate release.

Nine different ERjs reside in the human ER (Table 1) [79,80,81,130,145,146,147,209,210,211,212,213,214,215,216,217,218,219,220,221,222,223,224,225,226,227,228,229,230,231,232,233,234]. As the name infers, ERjs are characterized by individual J-domains, which allow interaction with BiP via the bottom of its NBD and, to do so, contain four α-helices (helices I–IV) with a loop region containing a highly conserved tripeptide of histidine, proline, and aspartic acid (HPD motif) located between helices II and III. They can be divided into membrane proteins with a lumenal J-domain and into bona fide lumenal proteins (Figure 5b). Only ERj3 through ERj6 appear to be involved in protein folding under physiological as well as ER stress conditions and in ERAD. The other ERjs play more specialized roles in ER protein import (Sec63/ERj2 and ERj1) [79,80,81,102,143,145,146,147,151] or ER-phagy (ERj8) [234]. Thus, there is redundancy also at the level of the ERjs, which may explain the non-lethal phenotype of loss of Sec63 function that is associated with polycystic liver disease (see below). Last but not least, Sec63 [102,143,155] and ERj3 together with ERj6 [131] act as BiP co-chaperones in facilitating conformational changes and, therefore, regulation of the Sec61 complex, which will be discussed in detail in the two following sections.

Grp170 does not only act as a Hsp70-type chaperone, it also acts as one of the NEFs for BiP [201,202,203,204]. There is another functional homolog to bacterial GrpE in the ER lumen (termed Sil1 or BAP) [200,204], i.e., there is redundancy also at the level of the NEFs, which may explain the non-lethal phenotype of loss of Sil1 function that is associated with the neurodegenerative disease, Marinesco–Sjögren syndrome (see below and Table 1). The structures of the two cytosolic paralogs of the two ER-resident NEFs were solved and revealed distinct interacting surfaces with the top of the amino-terminal nucleotide-binding domain (NBD) of the Hsp70 [205,206]. Thus, the NEF binding sites on the Hsp70 are different from the J-domain binding site. 

### 3.2. BiP and Sec62 as Allosteric Effectors of the Sec61 Complex for Channel Closing

The human ER represents the major Ca^2+^ storage compartment in nucleated cells and allows the controlled release of Ca^2+^ from the ER upon hormone stimulation of a resting cell, e.g., via IP3- or ryanodine receptors [5,6,7,8,9,10,11,12,13]. Subsequently, Ca^2+^ is pumped back into the ER by SERCA to re-establish the steep and crucial ER to cytosol Ca^2+^ gradient [8]. In addition, this gradient is constantly challenged by the so-called passive Ca^2+^ efflux from the ER. Therefore, SERCA has the additional task of counteracting this Ca^2+^ leakage. Furthermore, Ca^2+^ is taken up by mitochondria. In the course of the last fifteen years, several proteins were linked to ER Ca^2+^ leakage, including the Sec61 channel [60,61,62,63,64,65,66,67,68,69]. Therefore, the Sec61 channel gating is tightly controlled (Figure 2 and Figure 5).

Originally, sophisticated biophysical measurements on ER-derived membranes established that closing of the aqueous Sec61 channel involves BiP and an unidentified JDP to preserve the ER membrane permeability barrier [197,198,207]. Subsequently, single-channel recordings from planar lipid bilayers characterized the Sec61 complex as a highly dynamic aqueous channel that is transiently opened by SPs and permeable to Ca^2+^ after completion of protein import [61]. The same experimental strategy showed that the Sec61 channel closes either spontaneously or as induced by binding of BiP or Ca^2+^-CaM [61,69,235]. The fact that BiP is involved in closing the Sec61 channel was confirmed at the cellular level by combination of siRNA-mediated gene silencing or pharmacological manipulation and live cell Ca^2+^ imaging [69]. In addition, cytosolic Ca^2+^-CaM was shown under similar conditions to contribute to Sec61 channel closing via an unrelated mechanism after Ca^2+^ has started to leak from the ER [68]. During the last ten years, additional siRNA-mediated gene silencing and live cell Ca^2+^ imaging experiments characterized the pair and, possibly, heterodimeric complex of ERj3 and ERj6 as specific co-chaperones of BiP and the putative EF hand- and Ca^2+^- binding protein Sec62 as a co-factor of CaM in Sec61 channel closure (Figure 2 and Figure 5b) [131,236]. The idea that a heterodimeric co-chaperone complex is involved was based on the observation that depletion of ERj6 resulted in a twofold overproduction of ERj3 but the increased ERj3 level did not compensate the decreased ERj6 level in limiting Ca^2+^ leakage. The binding site of BiP was identified as the above-mentioned di-tyrosine motif–containing mini-helix within ER lumenal loop 7 of the Sec61α [69] and was shown to be relevant to the described mechanisms in cellulo by mutagenesis studies. Again, the idea is that binding of BiP to loop 7 of Sec61α provides binding energy for shifting the dynamic equilibrium of the Sec61 channel to the closed state. In case of inefficient channel closure in intact cells, Ca^2+^ starts to leak from the ER into the cytosol and binds calmodulin, and Ca^2+^-CaM is recruited to the IQ motif in the Sec61α -subunit [68] (Figure 2 and Figure 5b). Once again, the involved binding energy favors channel closure. Apparently, binding of Ca^2+^-CaM is supported by Sec62, which may bind Ca^2+^ because of its putative EF hand within its cytosolic carboxy-terminal domain [236]. Next, the Sec61 channel is closed, and Ca^2+^ leakage subsides. SERCA pumps Ca^2+^ back into the ER, CaM and Sec62 return to the Ca^2+^-free forms, and the next protein import cycle can be initiated. When these mechanisms fail, however, the passive Ca^2+^ efflux of the ER membrane might actually represent part of a signaling pathway reporting about protein homeostasis and folding capacity within the ER lumen and, eventually leading to apoptosis.

## 4. Sec61-Channelopathies

As outlined above, the Sec61 channel of the human ER membrane and its allosteric effectors (TRAP, Sec62, Sec63, and BiP) plus BiP´s co-chaperones and NEFs play central roles in extra- and intra-cellular proteostasis as well as in intracellular Ca^2+^-homeostasis. Therefore, the term Sec61-channelopathies was coined for diseases, which are the result of toxin-driven or hereditary defects in one of the three Sec61 subunits themselves or in one of the many allosteric effectors of the Sec61 channel (Figure 6) [86]. In general, genetically-determined defects can affect a single or both alleles coding for a certain component and are termed heterozygous or homozygous; heterozygous mutations can result in haploinsufficiency, where the product of the wild type allele cannot compensate the loss of function of the mutated one, or in a dominant negative effect of the product of the mutated allele. Bacterial and fungal toxins can have similar effects as mutations [237]. On the other hand, some components of the interaction network of our interest here appear to have overlapping functions and, therefore, may be partially able to substitute for each other. In addition, some of the components were found to be overproduced in various types of tumor diseases, suggesting that overproduction and gain-of-function of a component can cause or support a disease state, too. Next, we summarize the current knowledge about these diseases, knowing the basic medical principle that “he who has fleas can also have lice” or, at the molecular level, that a certain disease may be the result of simultaneous lack of one function of the component and dominant negative effect of one of its additional functions.

It is noteworthy in the context of Sec61-channelopathies that endosome-resident Sec61 complexes were shown to be involved in antigen transport from endosomes into the cytosol for cross-presentation to CD8^+^ T cells [238]. In the so-called endosome to cytosol pathway of cross-presentation, antigens are exported from endosomes into the cytosol by endosome-resident Sec61 channels and degraded by the proteasome, in possible analogy to Sec61-dependent ERAD. Therefore, this moonlighting function of the Sec61 channel has always to be considered in Sec61-related diseases, as may equally be true for possible specialized functions of Sec61 channels in ERAD.

### 4.1. Bacterial and Fungal Toxins That Target the Sec61 Channel

During the last ten years an ever-growing number of small molecule Sec61 channel inhibitors was discovered in bacteria and fungi that are best discussed in light of the energetics and kinetics of Sec61 channel gating (Figure 2 and Figure 3) [239,240,241,242,243,244,245,246,247,248,249,250,251,252,253,254,255]. Some of these bacterial and fungal products are synthesized by pathogenic organisms and toxic to humans (Table 2, Figure 6). Mycolcatone from *Mycobacterium ulcerans* is central to the etiology of Buruli ulcer [242,245]. In general, these small molecules affect ER protein import at the level of the Sec61 channel in a either precursor-specific or non-selective manner. According to the kinetic point of view on Sec61 channel gating, inhibitor selectivity is based on the distinct efficiencies of different amino-terminal SPs and TMHs in reducing the activation energy for Sec61 channel opening and the common principle that the bound inhibitors increase the energy barrier for opening of the Sec61 channel. The first-described and precursor-selective class of such inhibitors were the cyclic heptadepsipeptides, i.e., the fungal product HUN-7293 and its synthetic relatives CAM749 and Cotransins (e.g., CT8) [239,240,243]. Subsequently, the structurally unrelated compounds Apratoxin A, Mycolactone, Coibamide A and Impomoeassin F were characterized as Sec61 effectors and shown to have selective (Mycolactone) or non-selective (Apratoxin A, Coibamide A, Ipomoeassin F) effects on ER protein import by interaction with the Sec61 channel [244,245,246,247,248,249,251,253]. Although the different bindings sites of these small molecules within the Sec61 channel have been characterized by the selection of resistant Sec61α variants and cryo-EM [243,249,251,252], respectively, the exact mechanisms of these compounds is an open question. In first attempts to address this puzzle, it was asked whether or not the selectivity of some of the small molecules correlates with the dependence of some precursors on allosteric Sec61 channel effectors and whether or not the inhibitory compounds affect cellular Ca^2+^ homeostasis. With respect to the first question it was observed in two independent studies that the import of the BiP- and Sec63-dependent precursors of proapelin and ERj3, respectively, into the human ER is sensitive to CAM741 [102,143], i.e., that the effect may indeed be related to SP strength. With respect to the second question the synthetic non-selective Sec61 inhibitor Eeyarestatin 24 (ES24) was found in human cells to trap the Sec61 channel in a partially open state, which allowed the passage of Ca^2+^ but not of precursor polypeptides, was termed “foot in the door” and may be identical with the primed state of the channel [241,250]. As a result, the compound induced Ca^2+^-dependent apoptosis. Recently, similar phenotypes were observed for Mycolactone (T. Pick, R. Simmonds, A. Cavalié, personal communication). Thus at least some of the Sec61 inhibitors have a dual effect on the channel, decrease of ER protein import and increase of Ca^2+^ leakage, a clear case of “he who has fleas can also have lice”. These first mechanistic experiments are all consistent with the kinetic view on Sec61 channel gating but, obviously, have to be extended to additional Sec61 inhibitors in future experiments.

Notably in this context, the Sec61 channel is also affected by a bacterial protein toxin, *Pseudomonas aeruginosa* Exotoxin A, which enters human cells by endocytosis and retrograde transport and inhibits ER export of immunogenic peptides as well as export of antigens from endosomes in cross-presentation. Apparently, the exotoxin even induces recruitment of Sec61 complexes to endosomes. Therefore, the pathogenic bacterium compromises the immune system of infected humans and can cause pneumonia or sepsis [238,254]. Exotoxin A binds near the Ca^2+^-calmodulin binding site to the amino-terminal tail of Sec61α and arrests the channel in the closed state, which does not even allow the passage of Ca^2+^ [255]. 

### 4.2. Mutated Variants of the Sec61 Channel

The archetype of genetically-determined Sec61-channelopathies in humans are diseases with mutations in one or both alleles of one of the three ubiquitously expressed SEC61 genes that have functional consequences in Sec61 channel gating (Table 3) (Figure 1d and Figure 6) [86]. Mammalian cells, which are highly active in protein secretion, termed professional secretory cells, may be particularly sensitive towards problems in Sec61 channel closure and, therefore, constantly on the verge to apoptosis. This has recently been seen in human patients associated with dominant negative effects in the course of i) autosomal dominant tubulointerstitial kidney disease (ADTKD) and glomerulocystic kidney disease in kidney cells with the Sec61α 1V67G- or Sec61α 1T185A-exchanges [256,257], ii) hypogammaglobulinemia or primary antibody deficiency (PAD) in plasma cells with the Sec61α 1V85D exchange [258], plus iii) autosomal dominant severe congenital neutropenia (ADSCN) in neutrophils with the Sec61α 1V67G- or Sec61α 1Q92R-exchanges [256,259] and associated with Diabetes mellitus for the β-cells of the mouse with the homozygous Sec61α1Y344H exchange [260]. The fact that ERj6 (DNAJC3) is involved in Sec61 channel closure and that its absence in human patients, too, causes Diabetes mellitus is in perfect line with this interpretation [261]. However, efficient Sec61 channel closure is clearly not the only problem in the archetype Sec61-channelopathies; reduced ER protein import due to reduced levels of functional Sec61 complexes, i.e., haploinsufficiency, certainly also contributes to the respective disease phenotypes. While the clinical and laboratory features of affected patients are well characterized, the detailed molecular mechanisms giving rise to the tissue- and organ-specific defects despite the ubiquitous expression of the SEC61A1 loci are still unclear.

#### 4.2.1. Sec61α 1 p.V67G and p.T185A in ADTKD

In three independent families with seven, two and one patient(s), respectively, suffering from autosomal dominant tubulointerstitial kidney disease (ADTKD) and glomerulocystic kidney disease with congenital anemia, respectively, heterozygous *SEC61A1* mutations were identified. These are two missense mutations causing the amino acid substitutions V67G (in the plug helix) and T185A (near the pore ring in TMH 5), respectively [256,257]. The T185A mutation caused ADTKD with a more severe and complex tubular phenotype. Two individuals in the one family with the V67G variant, however, also suffered from recurrent cutaneous abscesses requiring hospitalization until the age of 12 years, which was the result of neutropenia and is discussed next. Both variants were found to be delocalized to the Golgi apparatus after transient expression in HEK293 cells as well as in a renal biopsy from a patient. Replacement of wildtype Sec61α by either one of the two variants in Zebrafish embryos induced convolution defects of the pronephric tubules, which is consistent with tubular atrophy observed in the patients. Furthermore, immunohistochemical analysis detected the absence of staining for the secretory protein renin in juxtaglomerular granular cells from a patient with the T185A substitution but, instead, renin staining in the cytoplasm of tubular cells, representing a phenocopy of patients with mutations in the *REN* gene. This renin secretion deficiency may have been the result of the Sec61 haploinsufficiency and the major cause for the disease phenotype. Notably, the *MUC1*, *REN* and *UMOD* genes, which code for the precursors of mucin-1 (MIM: 174000), renin (MIM: 613092) and uromodulin (MIM: 191845), respectively were previously linked to ADTKD [262].

#### 4.2.2. Sec61α 1 p.Q92R and p.V67G in ADSCN

In three patients suffering from autosomal dominant severe congenital neutropenia (ADSCN) *SEC61A1* missense mutations were described for two independent families. The heterozygous *SEC61A1* mutations identified included two missense mutations causing the amino acid substitutions V67G (in the plug helix) and Q92R (in the lateral gate TMH 2), respectively [256,259]. The patient with the Sec61α V67G variant also suffered from ADTKD (see above). In contrast, kidneys were morphologically normal in the patient with the Sec61α Q92R variant and kidney function remained normal. Both mutations were observed to cause reduced cellular Sec61 levels due to protein instability and dysregulated Ca^2+^ homeostasis. When wildtype Sec61α was replaced with either the Sec61α V67G or the Q92R variant in HeLa cells, Sec61-dependent ER protein import was decreased, while TA protein biogenesis was not compromised. In addition, when wildtype Sec61α was replaced with the Q92R variant in myeloid leukemic HL-60 cells, Ca^2+^ leakage from the ER was increased and differentiation to CD11b^+^CD16^+^ cells was reduced, suggesting UPR dysregulation that was confirmed by single-cell analysis of primary bone marrow-derived myeloid precursors. In addition, in vitro differentiation of primary CD34^+^ cells phenocopied the UPR upregulation and recapitulated the clinical arrest in granulopoiesis. In vitro modeling of the two mutations suggested a mechanistic pathway of UPR upregulation and subsequent selective arrest of myeloid precursors. Notably, the *ELANE* and *JAGN1* genes, which code for the precursors of neutrophil elastase or NE (MIM: 202700) and the tetraspanning membrane protein jagunal homolog 1 (MIM: 616022), respectively, were previously linked to ADSCN. NE secretion deficiency or JAGN1 deficiency may have resulted from the Sec61 haploinsufficiency and may have contributed to the disease phenotype. Taken together with the observed increased Ca^2+^ leakage, the two mutated Sec61 channels represent additional cases of “he who has fleas can also have lice”.

#### 4.2.3. Sec61α 1 p.V85D and p.E381* in CVID

In two families with several patients suffering from early-onset, severe, recurrent bacterial infections in the respiratory tract and normal B- and T-cell subpopulations in the peripheral blood but immunoglobulin deficiencies involving IgM, IgG, and IgA were diagnosed [258]. Hence the diagnosis was primary antibody deficiency or PAD or, specifically, common variable immunodeficiency or CVID. After initiating immunoglobulin substitution therapy the patients have benefitted from a significant decrease in both number and severity of infections. Notably, clinical laboratory values that were found to be altered in patients with *SEC61A1*-linked ADTKD were normal in *SEC61A1*-linked CVID patients. In vitro stimulation of B cells from these patients showed a deficiency to develop and proliferate into plasma cell clones. The heterozygous *SEC61A1* mutations identified included the missense mutation V85D (i.e., the pore ring- and hydrophobic patch-residue in TMH 2) as well as one nonsense mutation introducing a premature stop at E381* (TMH 8), i.e., a truncated variant. When wildtype Sec61α was replaced with the Sec61α V85D variant in HeLa cells, Ca^2+^ leakage from the ER was increased (possibly due to a more polar pore ring) and Sec61-dependent ER protein import was decreased (possibly due to a less hydrophobic patch), while TA protein biogenesis was not compromised. Thus, the heterozygous CVID-linked missense variant can be expected to behave in a similar manner in patient cells, i.e., to show a reduced capacity for ER protein import and an increased Ca^2+^ efflux from the ER; the haploinsufficiency can also be excepted from the truncated E381* variant. Indeed, the immunoglobulin secretion capacity of plasma cells from both types of patients was reduced. Furthermore, in various multiple myeloma cell lines it was observed that Sec61α V85D over-production selectively impairs plasma cell lines and the patients did not suffer from deficiencies in other plasma proteins. In addition, in the same plasma cell lines the levels of the UPR sensors IRE1 and PERK were increased and the apoptosis-inducing CHOP was activated. Intriguingly, mimicking haploinsufficiency by reduction of the Sec61 levels in the plasma cell lines via siRNA treatment phenocopied over-production of the V85D variant. Thus, even Sec61 haploinsufficiency is not easily tolerated by plasma cell lines. Taken together, Sec61α1 p.V85D and p.E381* apparently cause CVID because of unresolvable ER stress in the course of differentiation of B cells to plasma cells during a bacterial infection. Since the dominant negative effect of the Sec61α V85D variant is stronger than the truncation, the patients with the missense mutation suffer from a fully penetrant CVID, while the single available patient with the truncation may show transient episodes of hypogammaglobinemia.

#### 4.2.4. Sec61α1 p.Y344H in Mice in Diabetes mellitus

A murine model for Diabetes mellitus has indicated that a point mutation in the ER lumenal loop 7 of murine Sec61α leads to a partially deficient Sec61 complex and to β-cell death and diabetes [260]. Interestingly, insulin secretion per se was not compromised by the mutation. However, when protein levels of ERj3 were analyzed in pancreas- and liver-tissue from homozygous Sec61α^+/+^ heterozygous Sec61α^+/Y344^ and homozygous Sec61α^Y344H/Y344H^ mice the presence of the mutated Sec61α was found to cause reduced ERj3 levels in both organs [143]. Thus, the BiP-dependence of ERj3 import into the mammalian ER and its action via Sec61α loop 7-interaction were confirmed in different tissues of adult mice. When wildtype Sec61α was replaced with the corresponding mutant Sec61α Y344H in HeLa cells, Ca^2+^ leakage from the ER was increased and was no longer affected by manipulation of the BiP concentration [69]. Therefore, it was suggested that failure of BiP to facilitate Sec61 channel closure in the homozygous *SEC61A1Y344H* mouse contributes to apoptosis of cells with high secretory activity, such as pancreatic β-cells. It is interesting to mention that various other mutations and knock-outs of resident ER proteins can cause diabetes in mice, such as deletion of BiP´s Hsp40-type co-chaperones ERj4 and ERj6 [222,225,233], or of the BiP-interacting protein PKR-like kinase (PERK) [263]. However, diabetes can also be caused in man and mice by mutations in genes coding for non-ER proteins, such as the insulin gene [88].

## 5. Diseases That Are Related to Allosteric Effectors of the Sec61 Channel

### 5.1. Loss of TRAP Function in Congenital Disorders of Glycosylation (CDG)

Congenital disorders of glycosylation (CDG) are typically autosomal recessive diseases, which are characterized by altered protein or lipid glycosylation and, as a result, neurological abnormalities. Various genes have been linked to the disease, most notably and not surprisingly several genes, which code for subunits of oligosaccharyltransferease that catalyzes the first step of *N*-glycosylation in glycoprotein synthesis (Table 1). Typically, the patients are first identified by the characterization of carbohydrate-deficient plasma proteins, such as transferrin. Recently, mutations in the human TRAPγ and TRAPδ subunits (SSR3 and SSR4, respectively) were found to result in loss of TRAP and congenital disorders of glycosylation (CDG) [56,143,264,265], suggesting that TRAP plays a direct or indirect role in the biogenesis of *N*-glycosylated proteins (MIM 300090 and 606213). 

As described above, the combination of siRNA mediated depletion of a certain transport component from human cells with subsequent cellular protein abundance analysis identified the precursors of *N*-glycosylated as well as unglycosylated secretory and membrane proteins as TRAP substrates and characterized their SPs with above-average glycine-plus-proline content and below-average hydrophobicity as distinguishing for TRAP dependence [143]. Furthermore, control fibroblasts and three CDG patient fibroblasts with TRAP-deficiency were subjected to label-free quantitative proteomic analysis plus differential protein abundance analysis and the data were analyzed for negatively affected proteins, i.e., potential TRAP substrates. The proteomic analysis confirmed the almost complete absence of all TRAP complex subunits in fibroblasts from CDG patients with mutations in the *TRAPG* or *TRAPD* genes and the absence of the OST subunit TUSC3. Furthermore, the analysis of these chronically TRAP-depleted cells partially confirmed that the glycine-proline-content of SP plays an important role for TRAP dependence of precursor polypeptides in ER protein import. Thus, TRAP plays a precursor specific role in ER protein import, including precursors of *N*-glycoproteins and membrane proteins.

Although not only glycoproteins were affected by TRAP-depletion in HeLa cells and human CDG patient fibroblasts, the quantitative proteomic results confirmed the *N*-glycosylation deficiency associated with TRAP deficiency. They suggested that this may result directly from the depletion of TRAP, or from its secondary effects on OST or from a defect in a potential supportive role of TRAP in *N*-glycosylation, which would not be unexpected in light of the direct interaction of TRAP and OST seen in CET (Figure 1c). In any case, CDG as a result of TRAP deficiency is a precursor specific ER protein import defect. The question why this defect is not lethal may best be explained with the kinetic model for Sec61 channel gating (Figure 3), the allosteric effector affects the kinetic but not the equilibrium of the reaction. Furthermore, Sec63 may be able to compensate loss of TRAP function for some precursor polypeptides.

### 5.2. Sec63 and Sec61β in Autosomal Dominant Polycystic Liver Disease (ADPCLD)

Various human organs can develop multiple cysts, i.e., fluid-filled sacs, which are, typically formed by a certain cell type [266,267,268]. The classical example of this type of human inherited disease is autosomal polycystic kidney disease (ADPKD; MIM 173900) [266,267]. It is characterized by multiple renal cysts as well as additional extra-renal manifestations, such as cysts in the liver bile ducts. Mutations in two genes were linked to ADPKD: *PKD1* codes for polycystin 1 (PC1), a plasma membrane resident receptor, and *PKD2* codes for polycystin 2 (PC2) or TRPP2, a member of the transient receptor potential protein family that resides in the ER and the plasma membrane. Some research on ADPKD suggested *PKD* mutations result in problems in downstream signaling- as well as cell adhesion components, most notably ß-catenin, and that these problems cause alterations in planar cell polarity and tubular morphogenesis that, eventually, result in cyst formation. 

Autosomal dominant polycystic liver disease (ADPLD or PCLD, MIM 174050) is an ADPKD-related human disorder, characterized by the progressive development of biliary epithelial liver cysts (Figure 7) [269,270,271,272,273]. This inherited disease usually remains asymptomatic at young ages and manifests between the ages of 40 and 60 years. Liver function is usually preserved. Although ADPLD patients hardly ever suffer from other polycystic organ disorders, the etiologies of ADPKD and ADPLD may be related. On the genetic level, ADPLD is heterogeneous involving at least five different genes, which code for proteins that are involved in the biogenesis of secretory and plasma membrane proteins (Table 1) [270,271,272,273]: (i) *ALG8*, which codes for alpha-1,3-glucosyltransferase that is involved in the dolichol-linked oligosaccharide precursor for N-linked glycosylation; (ii and iii) *GANAB* and *PRKCSH*, encoding the α- and β-subunit, respectively, of glucosidase II, which is a resident protein of the ER lumen and involved in the folding of glycoproteins [268], (iv and v) *SEC61B* and *SEC63*, which code for the β-subunit of the Sec61 complex and Sec63, respectively, that reside in the ER membrane and are involved in ER protein import. A loss of function was postulated for all cases described. Although no mechanism has been firmly established for ADPLD, the etiology of the disease is best explained by a two-hit mechanism. Accordingly, patients with one inherited mutant allele and one wild type allele may at some point lose the function of the second allele in a few cholangiocytes through somatic mutation. Next, the progeny of these cells develop into cysts. The mechanism of cyst development is still unclear. The most likely scenario would be that all five proteins are essential for the biogenesis of a single protein or a set of proteins that are involved in the control of biliary cell polarity or cell adhesion, and in the absence of their functions, these proteins do not reach their functional location. This could result in alterations in planar cell polarity and morphogenesis. This view was confirmed and it was concluded that the secondary lack of PC1 and PC2 results in disrupted cell adhesion and, therefore, cyst formation [155,272]. Notably, *ALG8* was also linked to CDG (see above). Depending on the severity of the disease, the treatment options for this disease extend from aspiration of cysts as guided by ultrasound or computer tomography, to liver resection, all the way to liver transplantation (Figure 7).

Following the depletion of Sec63 from HeLa cells, the quantitative proteomic results confirmed a precursor specific ER protein import defect, which affected the biogenesis of soluble and membrane proteins irrespectively of whether they were *N*-glycosylated or not. An additional Sec63 function has to be considered. Sec63 also plays a role as nucleoredoxin interactor and, therefore, may be involved in the Wnt/ β-catenin signaling pathway, which also has been shown to play a role in planar cell polarity. However, the fact that three additional proteins, involved in protein biogenesis at the ER but involved in entirely different aspects of this process, cause the same disease argues strongly for the interpretation, that the ER import defect caused by Sec63 deficiency is responsible for the disease. The question why this defect is not lethal may best be explained with the kinetic model for Sec61 channel gating (Figure 3), but may also have something to do with ERj1, which may have overlapping functions with Sec63. 

### 5.3. BiP and Its Co-Chaperones and NEFs in Diabetes and Neurological Disorders

#### 5.3.1. BiP Deficiency in Hemolytic Uremic Syndrome (HUS)

Shiga toxigenic *Escherichia coli* (STEC) strains can cause morbidity and mortality in infected humans [274]. Some of these pathogens produce amongst various others AB_5_ toxin or subtilase AB (SubAB) and are responsible for gastrointestinal diseases, including the life-threatening haemolytic uraemic syndrome (HUS) (MIM 235400). During an infection, the bacterial cytotoxin enters human cells by endocytosis and retrograde transport delivers it to the ER. In the ER, BiP is the major, or possibly the only, target of the catalytic subunit A, which inactivates BiP by a single reaction of limited proteolysis within the linker region (Figure 5b). Eventually, all above-outlined BiP functions are lost, and the affected cells die. Therefore, global loss of BiP function is not compatible with life. This acquired disease is by definition a chaperonopathy, i.e., the result of a certain chaperone deficiency. Here, it is mentioned in the context of Sec61 channelopathies for comparison.

#### 5.3.2. Marinesco–Sjögren Syndrome (MSS)

Mutant variants of BiP interaction partners are associated with the manifestation of neurological diseases [275,276,277,278,279,280,281,282,283,284,285,286,287,288,289,290,291,292,293,294,295,296,297,298,299,300,301,302,303,304,305]. In 2005, recessive *SIL1* mutations were linked to the phenotypical manifestation of Marinesco–Sjögren syndrome (MSS; MIM: 248800) (Figure 8), a rare autosomal recessively inherited multisystemic disorder characterized by a vacuolar myopathy, congenital or infantile manifesting cataracts and cerebellar atrophy leading to ataxia (Figure 8b) [276,277]. Intellectual disability occurs in the majority but not in all patients [280]. Moreover, a vulnerability of the peripheral nervous system in terms of axonal degeneration was identified. A mouse model of the disease called ‘woozy´ represents a good phenocopy of the human disease by showing cerebellar atrophy characterized by a degeneration of Purkinje cells of the vestibulocerebellum leading to ataxia, a vacuolar myopathy as well as degeneration of peripheral axons (Figure 8c) [278,279,281,284,288]. Ultrastructural investigations of skeletal muscle of man and mouse showed a profound disintegration of the nuclear envelope characterized by the proliferation of the lamina fibrosa, a finding which is in line with the enrichment of BiP within the nuclear envelope in muscle cells (Figure 8a) [275]. Of note, mitochondrial degeneration is also a major ultrastructural finding in *SIL1/Sil1*-mutant muscle.

*SIL1* encodes the ubiquitously expressed Sil1 or BAP which is also controlled by ER-stress and induction of the unfolded protein response [287]. Quantitative studies of BiP, GRP170 and Sil1 in human muscle cells (RCMH) revealed a molecular ratio of 1:0.1:0.001 [286]. Beside the interaction with BiP, a binding to POC1A, a protein linking centrosomes to Golgi assembly was demonstrated [278]. Notably, pathogenic missense variants of Sil1 lead to a disruption of the SIL1-POC1A interaction which is in turn associated with centrosome disintegration [289]. Further morphological and biochemical studies on an in vitro model (Sil1-depleted HEK293 cells) utilizing electron microscopy and unbiased proteomic profiling revealed structural changes of the ER including the nuclear envelope and mitochondrial degeneration that closely mimic pathological alterations in MSS as well as indicated that proteins involved in cytoskeletal organization, vesicular transport, mitochondrial function, and neurological processes contribute to Sil1 pathophysiology [285]. Moreover, a particular function of Sil1 for etiopathology of two neurodegenerative disorders, amyotrophic lateral sclerosis (ALS) and Alzheimer disease was highlighted, thus declaring the functional Sil1-BiP complex as a modifier for neurodegenerative disorders [282,283,287].

The hypoxia up-regulated protein 1 (HYOU1/Grp170) displays an BiP-independent chaperone activity but represents another NEF for BiP, too. This raises the question why loss of functional Sil1 cannot be compensated by increased expression of *HYOU1*: Previous studies have shown that over-expression of Grp170 in mice causes severe myopathic changes of skeletal and cardiac muscle. Thus, forced expression of Grp170 might result in a worsening of the muscular pathology rather than in an amelioration of the pathology. However, focussing on the nervous system in the “woozy” mouse model, Zhao and co-workers demonstrated that overexpression of *HYOU1* prevents ER stress and rescues neurodegeneration, whereas decreasing expression of *HYOU1* exacerbates these phenotypes [290,291]. Hence, one might assume that different tissues show varying tolerances against the increased expression of *HYOU1*.

ERj6/DNAJC3/p58(IPK) is a co-chaperone that promotes ATP hydrolysis by BiP and is involved in folding, assembly, ERAD and Sec61 gating to the closed state (the latter together with ERj3). In vivo studies utilizing the “woozy” mouse model revealed that decrease of ERj6 ameliorates ER stress and neurodegeneration in these animals suggesting that alterations in the nucleotide exchange reaction of BiP cause ER stress and neurodegeneration in Sil1-deficient neurons [278,279]. In 2014, recessive *DNAJC3* mutations were linked to Diabetes mellitus complicated by multisystemic neurodegeneration including ataxia, upper-motor-neuron damage, peripheral neuropathy, sensorineural hearing loss, and cerebral atrophy [261] (MIM: 616192) and a further clinical and molecular genetic study confirmed the phenotype associated with recessive loss of function mutations within *DNAJC3* [292]. Although precise molecular data unravelling the underlying pathomechanisms are still scarce a recent study points at BIM- and PUMA-dependent activation of the mitochondrial pathway of apoptosis [292]. 

Matrin-3 (MATR3) is a highly conserved phosphoprotein resident within the nuclear envelope playing a role in transcription. The interaction with other nuclear envelope/matrix proteins moreover suggests a function in the maintenance of the internal fibrogranular network. A study published in 2014 described an interaction with BiP [294], a molecular finding which accords with our observed co-localization of both proteins within the nuclear envelope (see above) as well as the identified dysregulation of MATR3 in the diseased muscle of “woozy” animals in terms a pathological proliferation of the lamina fibrosa. Functional data showed that downregulation of BiP triggers the caspase cascade pathway leading to MATR3 degradation [294].

Inositol 5-phosphatase (INPP5K/SKIP) acts on the inositol 1,4,5-trisphosphate, the inositol 1,3,4,5-tetrakisphosphate, the phosphatidylinositol 4,5-bisphosphate and the phosphatidylinositol 3,4,5-trisphosphate pathway [298]. In agreement with a described interaction with BiP [298], INPP5K localizes in part to the sarcoplasmic/endoplasmic reticulum where it is preferentially localized in ER tubules and enriched (relative to other ER resident proteins such as Sec61β), in newly formed tubules [300]. In 2017, recessive *INPP5K* mutations were linked to congenital muscular dystrophy with cataracts and mild cognitive impairment [301,302] (MIM: 617404). Like in MSS-patients, muscle pathology in INPP5K-patients is also characterized by the disruption of the architecture of the nuclear envelope with proliferation of the lamina fibrosa [302], suggesting common pathomechanisms. Thus, as for the pathophysiology related to MSS and the Matrin-3 associated phenotypes, also for INPP5K a defective protein clearance machinery seems to be one of the pathomechanisms contributing to the clinical manifestation of the diseases, especially in terms of the muscular phenotypes.

#### 5.3.3. ERj3 in Polycystic Kidney Disease (PKD)

Besides the classical autosomal polycystic kidney disease (ADPKD), which is characterized by multiple cysts and organ enlargement, there is a late-onset form of the disease (age 60+), which does not result in enlarged kidneys and shows progressive interstitial fibrosis, i.e., phenotypic overlap with autosomal-dominant tubulointerstitial kidney disease (ADTKD), which can result from certain *SEC61A1* mutations (see above). Recently, this PKD form was linked to the *DNAJB11* gene, which codes for ERj3, a co-chaperone of BiP that promotes ATP hydrolysis by BiP and is involved in folding, assembly, Sec61 gating to the closed state and ERAD (Table 1). Specifically, two mutations in the J-domain, p.P54R (affecting the HPD motif) and p.L77P, two frameshift mutations in the cysteine-rich domain and one truncation, respectively, were described [306]. In respective null cells of the gene and in kidney samples from affected individuals the pathogenesis was associated with secondary defects in PC1, the plasma membrane resident receptor that plays a role in cell polarity, and uromodulin, one of the most abundant secretory proteins in our body and the most abundant urinary protein. Mutant variants have previously been linked to the *UMOD* gene, which codes for the precursor of uromodulin (MIM: 191845). For the latter disease it has been suggested that mis-folded mutant variants accumulate in the ER of thick ascendant limb of the loop of Henle- or TAL-cells and cause progressive cell damage. Notably, however, ERj3 (DNAJB11) is involved in Sec61 channel closure and its absence in human patients, too, may cause additional problems in Ca^2+^-homeostasis.

## 6. Tumor Diseases That Are Related to the Sec61 Channel

Professional secretory cells with their abundant rough ER appear to be particularly sensitive to imbalances in the Sec62 to Sec63 ratio, which result in over-efficient Sec61 channel closure at a higher than average ratio and, thus, a proliferative and/or migratory advantage that can lead to cancer, e.g., seen after over-expression of *SEC62* in prostate or lung cancer. That may be due to the role of Sec62 in maintenance of Ca^2+^ homeostasis but may also be related to its role in ER proteostasis, i.e., ER-phagy, or both. Due to poor vascularization and the resulting hypoxia and glucose starvation, tumor cells are prone to ER stress and, therefore, UPR [307,308]. In cultured cells, BiP is one of the proteins involved in protecting cancer cells against ER stress-induced apoptosis [309]. In addition to this general link between BiP and cancer, some of the above-mentioned directly or indirectly interacting proteins of BiP have been connected to certain tumors.

Over the past 15 years, increasing evidence suggests a relevant role of *SEC61*, *SEC62,* and *SEC63* genes in the development and tumor cell biology of human malignancies (Figure 9) [310,311,312,313,314,315,316,317,318,319,320,321,322,323,324,325,326,327,328,329,330,331,332,333,334,335,336,337,338,339,340,341]. In 2002, a first publication described frameshift mutations of *SEC63* due to microsatellite instability (MSI) in 37.5% of gastric cancers and 48.8% of colorectal cancers [310]. Similar results were reported in 2005 [311] and in 2013 [324], where MSI associated *SEC63* frameshift mutations were found in 56% of small-bowel cancers in patients with hereditary non-polyposis colorectal cancer (HNPCC) [311] and in one case of hepatocellular carcinoma [324].

In recombinant inbred mouse lines showing different basal *SEC63* expression levels, low hepatic expression correlated significantly with a decrease in apoptosis and increased proliferative activity [324]. Taken together, the aforementioned studies suggest a function of *SEC63* as tumor suppressor gene in gastrointestinal and hepatic cancer, though the underlying molecular mechanisms largely remain unknown.

For the *SEC61* genes increased expression and gene amplification were reported for gastric cancer, colorectal cancer, alcohol-induced hepatocellular cancer, breast cancer and glioblastoma [314,315,316,317,323,330,336,337,338]. In glioblastoma multiforme (GBM), Liu et al. demonstrated a significant correlation of high *SEC61G* expression with poor prognosis based on statistical analysis of sequencing data from the Cancer Genome Atlas cohort (TCGA) and the Chinese Glioma Genome Atlas cohort (CGGA) [337]. Univariate and multivariate Cox proportional hazards regression verified *SEC61G* as an independent prognostic factor for prognosis and therapeutic outcome in these cohorts. Gene set enrichment analysis (GSEA) and gene set variation analysis (GSVA) suggested a connection to the Notch pathway as a possible molecular backbone for these observations, though more data are needed to prove this hypothesis. In view of potential therapeutic implications of these observations, Sec61 inhibitors including Exotoxin A, Mycolactone, Apratoxin A, Cotransin, and Eeyarestatin I were identified and proved to sufficiently block the translocation of precursor proteins as well as Ca^2+^ leakage through the Sec61 channel [331]. However, only Exotoxin A has been tested in first clinical studies on human cancer patients so far and it is not more than speculative if the role of Sec61 in ER protein import and/or Ca^2+^ homeostasis is responsible for its observed association with the clinical course of human cancer diseases. 

Highest evidence for a causative role of a protein translocation component in the development and tumor cell biology of human cancer exists for the ER transmembrane protein Sec62. In 2006, a first study found *SEC62* copy number gains in 7 of 13 prostate cancer samples as well as elevated Sec62 protein levels in three prostate cancer cell lines [313]. In the following years, amplification and overexpression of the *SEC62* gene was reported for various other cancer entities, including non-small cell lung cancer [318,320,328], thyroid cancer [318,320], hepatocellular cancer [322,340], ovarian cancer [325], breast cancer [325,334], head and neck squamous cell carcinoma [329,330], cervical cancer [336], vulvar cancer [335], atypical fibroxanthoma [339], and in larger prostate cancer patient cohorts [319]. When screening publicly available DNA sequencing data from over 72,000 cancer patients with 55 different tumor entities (cBio Portal for cancer genomics; https://www.cbioportal.org accessed on 21 April 2021) *SEC62* gene alterations are reported for 2595 patients and represent gene amplifications in the majority of cases (Figure 10a). 

However, from a functional point of view only few studies addressed the specific impact of altered *SEC62* expression levels on cancer cell biology. A first step to uncover potential associations of *SEC62* overexpression with tumor cell biology are correlation analyses with clinical data. Thereby, Greiner et al. found an association of high *SEC62* expression level with higher Gleason Score in prostate cancer [319]. In non-small cell lung cancer, high Sec62 levels correlated with the occurrence of lymph node metastases and poor tumor differentiation [320]. Similarly, an association of *SEC62* overexpression with lymphatic metastasis was reported for head and neck squamous cell carcinoma [330] as well as an association of *SEC62* overexpression with distant metastasis in breast cancer [334]. These results indicated a potential role of Sec62 in cancer metastasis, which was strengthened by several functional studies. A significant inhibition of cancer cell migration by *SEC62* gene silencing was reported for prostate cancer cells [318], non-small cell lung cancer cells [320], thyroid carcinoma cells [320], cervical cancer cells [328], hepatocellular carcinoma cells [340], and head and neck squamous cell carcinoma cells [330]. On the contrary, *SEC62* overexpression stimulated migration of HEK293, HeLa, Huh-7 and FaDu cells [236,319,330,340] and induced subcutaneous tumor growth in C.Cg/AnNTac-Foxn1^nu/nu^ mice, inoculated with *SEC62* overexpressing HMLE cells [325]. An influence of *SEC62* expression level on stress tolerance of human cancer cells was suggested by several studies reporting a higher sensitivity to ER stress induced by CaM inhibitors when *SEC62* is expressed at low levels [319,320,328]. Thereby, the regulation of Sec61-mediated Ca^2+^ efflux through the ER membrane by Sec62 is supposedly the key function for the migration stimulating effect and improved stress tolerance [328]. However, the broad influence of a potentially dysregulated ER-phagy in *SEC62* overexpressing cancer cells cannot be ruled out [32].

With regard to a potential prognostic relevance of *SEC62* expression, several studies reported a significant correlation of elevated Sec62 levels with poor patient prognosis in non-small cell lung cancer [328], breast cancer [334], liver cancer [322], and head and neck cancer [329,330]. *SEC62* expression was also identified as a potentially independent prognostic factor for early recurrence in postoperative HCC patients [340]. Consistently, a correlation of *SEC62* amplification with overall survival in 40,006 cancer patients (45 different tumor entities) based on publicly available DNA sequencing and clinical data (cBio Portal for cancer genomics; https://www.cbioportal.org, accessed on 21 April 2021) shows a highly significant association of *SEC62* amplifications with poor prognosis (Figure 10b). 

Together, these data strongly indicate a role of *SEC62* as a driver oncogene in various human cancers with a consistent association with poor prognosis, lymph node as well as distant metastasis and stress tolerance, which turns the Sec62 protein into an attractive target for anticancer therapy. As Sec62 is hardly accessible for monoclonal antibodies due to its intracellular location, alternative strategies had to be developed to achieve at least a functional knock-down. Thereby, based on the role of Sec62 in the regulation of Ca^2+^ efflux through the Sec61 channel, CaM inhibitors (e.g., Trifluoperazine, TFP) and inhibitors of SERCA (e.g., Thapsigargin, TG) were investigated as potential therapeutics. Indeed, CaM inhibitors showed a functional Sec62 knockdown by blocking Ca^2+^ efflux from the ER lumen [328] and inhibiting cancer cell migration in cervix and prostate cancer cells with inhibition of cancer cell proliferation at higher doses [328]. One first in vivo study reported a significant inhibition of seeding and growth of a subcutaneously injected head and neck squamous cell carcinoma cell line (FaDu) in BALB/cAnNRj-Foxn1^nu/nu^ mice by single and combined treatment with TFP and TG [333]. Ongoing in vivo studies focusing on lymphatic and hematogeneous metastasis have to show if the migration inhibition that was found for various cancer cell entities in vitro manifests as a clinically relevant phenotype in a living organism. Within the scope of such therapeutic concepts, an additional benefit of autophagy inhibitors such as bafilomycin A1 and chloroquine seems to be conceivable was well, due to the central function of Sec62 in the process of ER-phagy [332].

## Figures and Tables

**Figure 1 cells-10-01036-f001:**
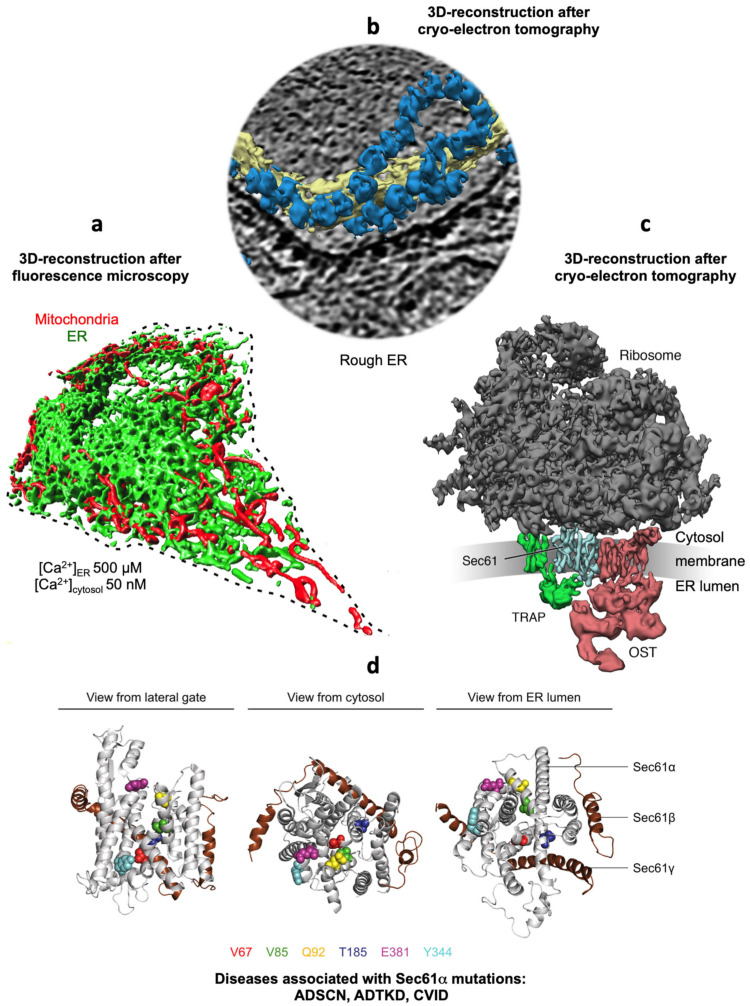
Collage of 3D reconstructions of a nucleated mammalian cell, a section of rough ER in such a cell and a ribosome-bound Sec61 translocon. (**a**) Represents a 3D reconstruction after live cell fluorescence imaging, following import of GFP into the ER and of RFP into the mitochondria. The plasma membrane is indicated by a dashed line. Typical concentrations of free Ca^2+^ are given for cytosol and ER of a resting cell. (**b**) depicts a 3D reconstruction of cellular rough ER after CET of a slice through the respective tomogram. ER membranes are shown in yellow; 80S ribosomes are shown in blue. (**c**) represents a 3D reconstruction of the native ribosome-translocon complex in rough microsomes. Here, the membrane density was removed to highlight membrane integral parts of the translocon complex. TMHs for Sec61 complex, TRAP and OST can be distinguished [56]. Helix 51 of an rRNA expansion segment and ribosomal protein eL38 represent the contact sites of TRAPγ, but are hidden by other ribosomal densities. (**d**) shows the PDB 3j7q structures for the Sec61 channel as seen from the indicated positions; disease associated point mutations are indicated. The collage is based on [57,58]. See text for details.

**Figure 2 cells-10-01036-f002:**
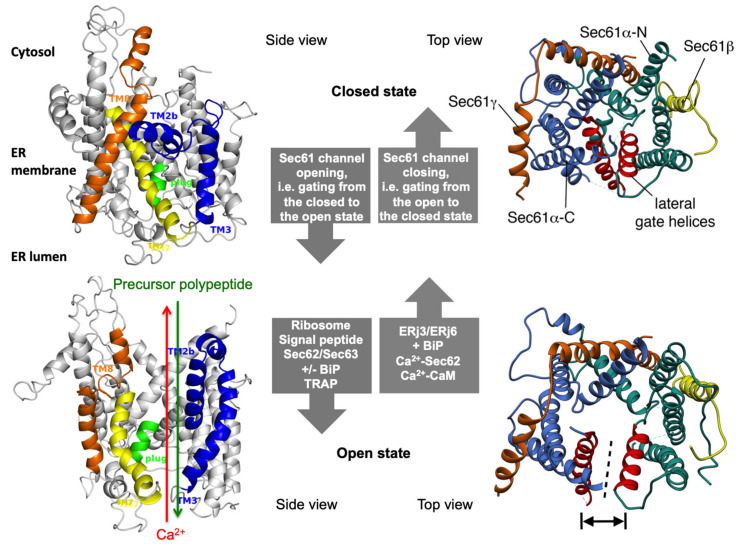
The concept of gating of the heterotrimeric Sec61 complex by signal peptides and allosteric effectors. The Sec61 channel is shown in its modeled closed (top) and open (bottom) conformational states, as viewed from the plane of the membrane (left) and in atomic models (PDB 3j7q, PDB 3jc2) as seen from the cytosol (right), respectively [57,58]. These two states are proposed to be in a dynamic equilibrium with each other. The fully open state of the Sec61 channel allows the initial entry of precursor polypeptides from the cytosol into the ER lumen and ER membrane, respectively, and is experimentally observed as cleavage of signal peptides by signal peptidase on the luminal side of the ER membrane. In addition, it allows the passive efflux of Ca^2+^ from the ER lumen into the cytosol and is visible in live cell Ca^2+^ imaging in cytosol and ER lumen. Ca^2+^ efflux may also be possible in the expected transition state (not shown), which may be identical to the so-called primed state that can be induced by ribosomes in co-translational- and by the Sec62/Sec63 complex in post-translational-transport. The conformational changes of the modelled Sec61 complex were previously morphed and the role of BiP plus an ERj co-chaperone, such as Sec63 and ERj1, respectively, visualized for co-translational transport at (see Data Availability). The Ca^2+^-permeability of the open Sec61 channel as observed by live cell Ca^2+^ imaging can be seen in the video file (see Data Availability).

**Figure 3 cells-10-01036-f003:**
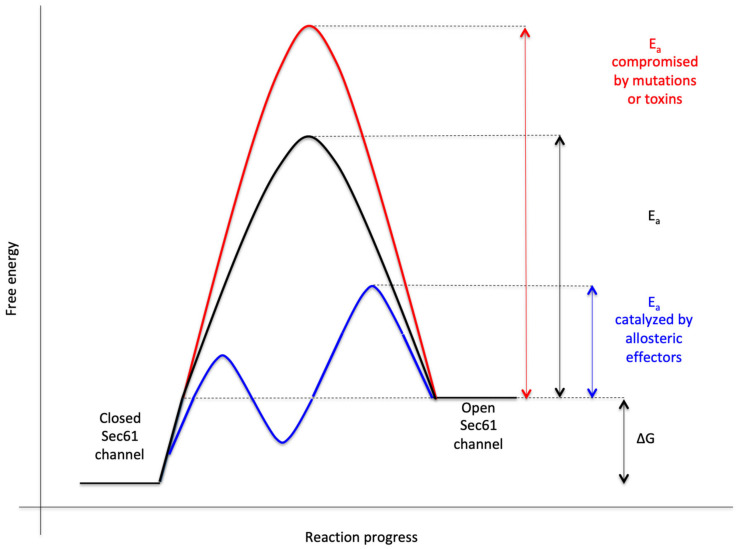
Energetics and kinetics of Sec61 channel gating. In our view, the TRAP− or Sec62/Sec63 +/− BiP-mediated Sec61 channel gating is best considered in analogy to an enzyme-catalysed reaction. Accordingly, TRAP, Sec62, Sec63 or BiP reduce the energetic barrier for full channel opening, which can apparently be reinforced by Sec61 channel inhibitors, such as cyclic heptadepsipeptides (such as CAM741) or certain eeyarestatins (such as ES24). At least in the case of ES24, binding of the inhibitor within the channel pore arrests the channel in a partially open state (termed “foot in the door”), which maybe identical with the primed state and is compatible with Ca^2+^-efflux but not with full channel opening for protein translocation. TRAP and BiP contribute to full channel opening by direct interaction with ER luminal loops 5 and 7, respectively, of Sec61α (see below). *SEC61A1* mutations can increase the energy barrier for channel opening per se (V67G, V85D, and Q92R mutation) or indirectly, such as by interfering with BiP binding (Y344H mutation). Notably, all these effects are precursor specific because the amino-terminal SPs are either efficient or inefficient in driving Sec61 channel opening. Typical for an enzyme-catalysed reaction, BiP can also support efficient gating of the Sec61 channel to the closed state, i.e., the reverse reaction.

**Figure 4 cells-10-01036-f004:**
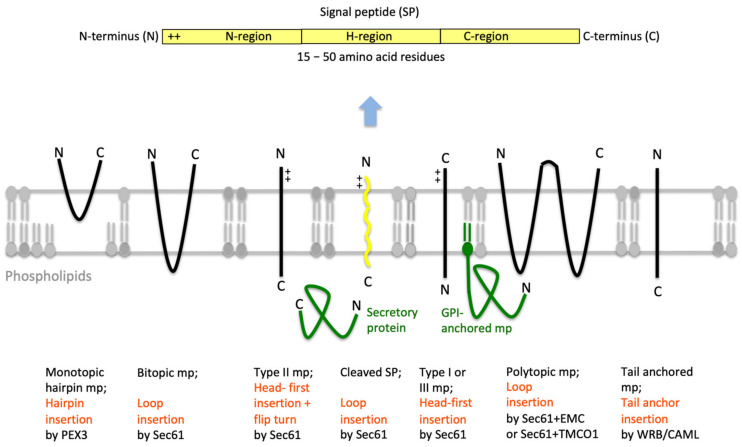
Features of amino-terminal signal peptides and ER membrane proteins. The cartoon depicts the signal peptides (SPs, shown in yellow) and six types of ER membrane proteins (in black), together with their membrane protein type and mechanism of membrane insertion (both indicated below the cartoon). Cleavable SPs (in yellow) can facilitate ER import of secretory proteins (in green), glycosylphosphatidylinositol (GPI)-anchored membrane proteins (in green) and various types of membrane proteins, except for hairpin-, type II- and tail anchored-membrane proteins. Positively charged amino acid residues (+) play an important role in membrane protein and SP orientation, i.e. typically, follow the positive inside rule. Bitopic and polytopic proteins can also involve SPs and have the opposite instead of the shown orientation. Alternatively, amino-terminal transmembrane helices (TMHs), which serve as SPs, facilitate membrane insertion. The shown bitopic protein is also named double-spanning membrane protein, the example polytopic protein is also named tetra-spanning membrane protein, if the shown type I membrane protein did not involve a cleavable SP it is also defined as signal anchor protein. In the case of membrane proteins with amino-terminal TMHs, membrane insertion typically involves the same components and mechanisms, which deliver secretory proteins (in green) and glycosylphosphatidylinositol (GPI)-anchored membrane proteins (in green) to the ER lumen. In certain cases, however, auxiliary membrane protein insertases, such as EMC or TMCO1 complex play a role. The latter two membrane protein complexes can also operate as stand-alone membrane protein insertases, an activity they have in common with the PEX- and the TRC-systems. Following their ER import, GPI-anchored membrane proteins become membrane anchored via their carboxy-termini by GPI-attachment. C, carboxy-terminus; N, amino-terminus.

**Figure 6 cells-10-01036-f006:**
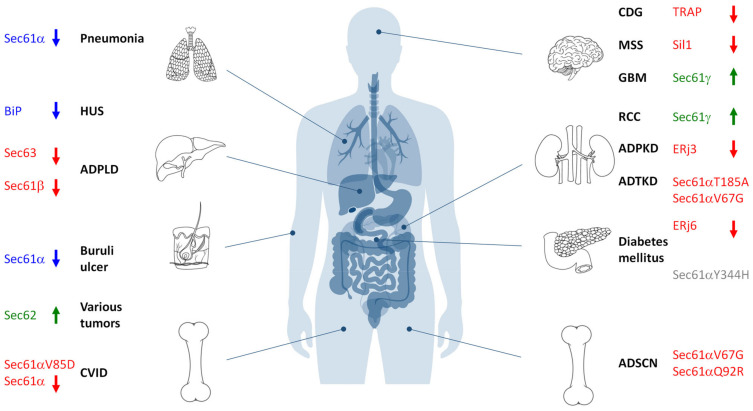
Hereditary and acquired diseases that are linked to the Sec61 complex and its allosteric effectors. The figure highlights various disease phenotypes, which are discussed in the text. Proteins affected in human hereditary diseases are indicated in red, protein targets of toxin-determined human diseases in blue, over-produced proteins in human tumor diseases in green (see below) and a genetically-determined variant causing murine Diabetes mellitus in grey. The arrows point upwards for increased activity of the indicated component and downwards for decreased activity. ADPKD, autosomal dominant polycystic kidney disease; ADPLD, autosomal dominant polycystic liver disease; ADSCN, autosomal dominant severe congenital neutropenia; ADTKD, autosomal dominant tubulo-interstitial kidney disease; CDG, congenital disorder of glycosylation; CVID, common variable immunodeficiency; GBM, glioblastoma multiforme; HUS, hemolytic uremic syndrome; MSS, Marinesco–Sjögren syndrome; RCC, renal cell carcinoma.

**Figure 7 cells-10-01036-f007:**
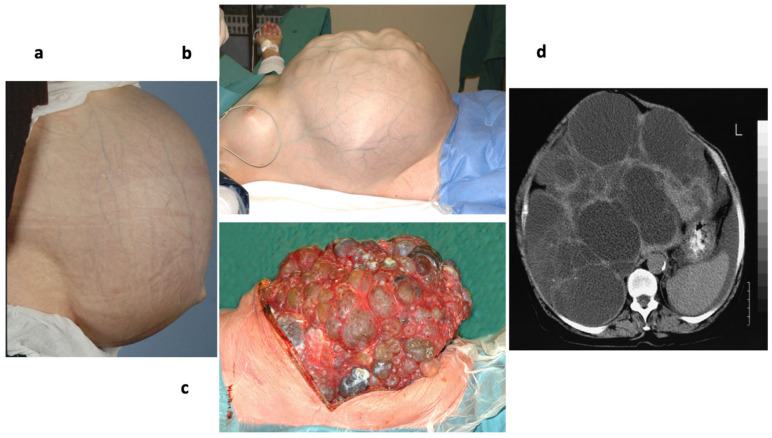
Autosomal dominant polycystic liver disease. Photographs of various ADPLD patients are shown in (**a**–**c**). (**d**) represents a computer tomogram of another patient.

**Figure 8 cells-10-01036-f008:**
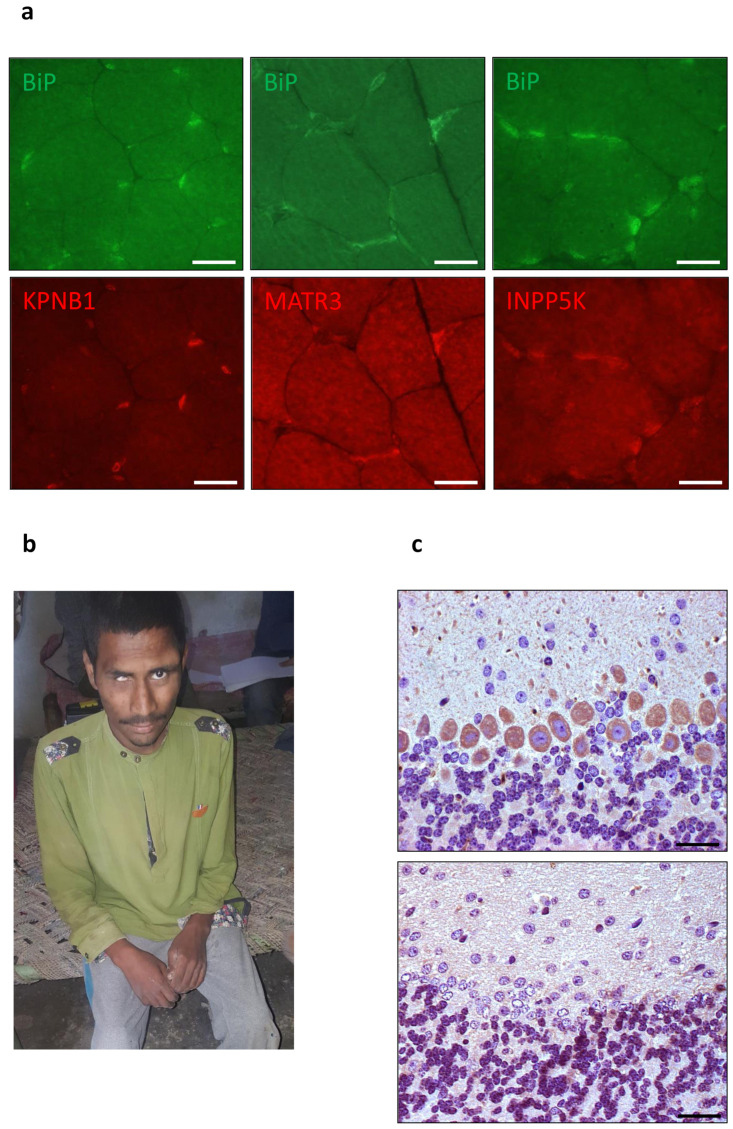
Marinesco–Sjögren-Syndrome. (**a**) BiP-enrichment in the nuclear envelope of muscle cells visualized by co-immunofluorescence on a human quadriceps muscle biopsy from a healthy donor; left panel highlights co-localization of BiP with KPNB1 (Importin subunit beta-1) within the nuclear envelope, the middle panel illustrates co-localization of BiP with Matrin-3, the right panel illustrates a co-localization of BiP with INPP5K. (**b**) A photograph of a genetically confirmed MSS patient from Pakistan. (**c**) Upper figure shows the cerebellum of a 26-weeks old wildtype mouse with regular appearance and distribution of Purkinje cells immunoreactive for STIM1 (brown labelling: Visualized by immunohistochemistry). Lower figure shows the cerebellum of a 26-weeks old woozy animal (no expression of Sil1 as a BiP co-chaperone) with loss of Purkinje cells. Scale bars: 50 µm.

**Figure 9 cells-10-01036-f009:**
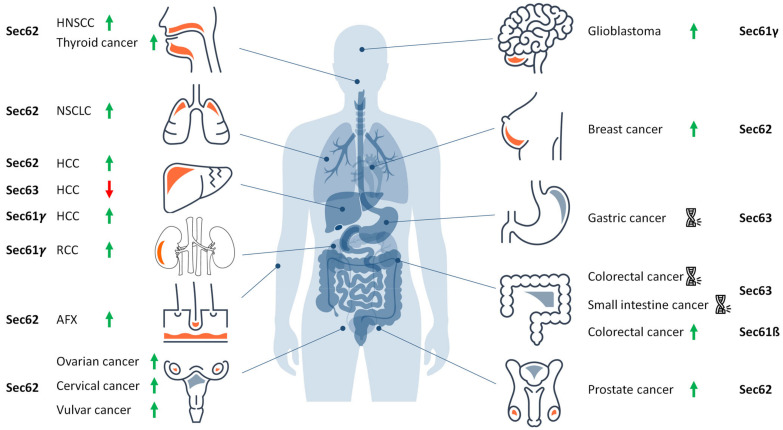
*SEC61*, *SEC62,* and *SEC63* in human cancer. Overview of genetic changes and altered expression of *SEC61*, *SEC62*, and *SEC63* gene in human cancer entities segregated by the tissue of origin (column-wise from top left to bottom right: Head and neck, lung, liver, kidneys, skin, female genital tract, brain, breast, stomach, intestine, and male genital tract). Green arrows symbolize functional gain by overexpression/amplification, red arrows symbolize functional loss by low expression/deletion, mutations are indicated by single-strand DNA break symbols. AFX—atypical fibroxanthoma; HCC—hepatocellular carcinoma; HNSCC—head and neck squamous cell carcinoma; NSCLC—non-small cell lung cancer; RCC—renal cell carcinoma.

**Figure 10 cells-10-01036-f010:**
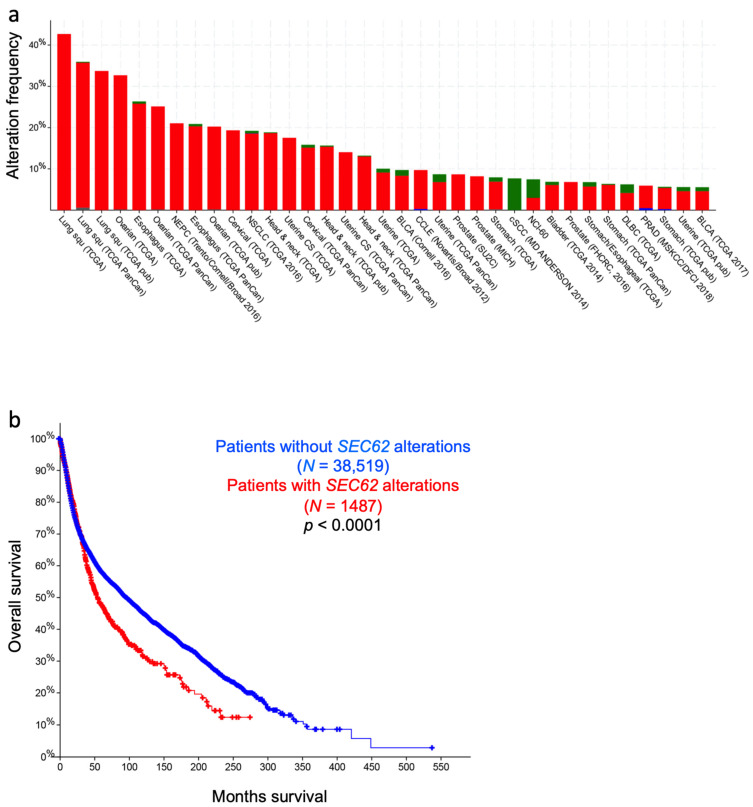
Genetic alterations of the *SEC62* gene and impact on overall survival. (**a**) Alteration frequency for the *SEC62* gene in a total cohort of 72,012 cancer patients based on publicly available DNA sequencing data entities (cBio Portal for cancer genomics). Results are illustrated only for a subset of patient cohorts with the highest alteration frequency. Red bars indicate gene amplification, green bars indicate gene mutation, blue bars indicate deep deletions. (**b**) Overall survival for patients with (red) and without (blue) alterations in the *SEC62* gene independent of alteration type. Censored data are labeled by crosses. In total, 40,006 patients were included in the survival analysis; 1487 patients showed *SEC62* alterations, 38,519 patients showed no *SEC62* alterations; median survival with *SEC62* alterations: 54.2 months, median survival without *SEC62* alterations: 95.6 months.

**Table 1 cells-10-01036-t001:** Protein transport components/*complexes* and associated proteins in HeLa cells.

Component/subunit	Abundance	Location	Linked Diseases
Calmodulin	9428	C	
*Cytosolic Chaperones*		C	
Hsc70 (HSPA8)	3559		
Hdj2 (DNAJA1)	660		
Bag1 (HAP, RAP46)	46		
*#NAC*		C	
- NACα	1412		
- NACβ			
*#SRP*		C	
- SRP72	355		Aplasia, Myelodysplasia
- SRP68	197		
- SRP54	228		Neutropenia, Pancreas Insufficiency
- SRP19	33		
- SRP14	4295		
- SRP9	3436		
- 7SL RNA			
*SRP receptor*		ERM	
- SRα (docking protein)	249		
- SRβ	173		
hSnd1	unknown		
*Snd receptor*		ERM	
- hSnd2 (TMEM208)	81		
- hSnd3	49		
*#Bag6 complex*		C	
- TRC35 (Get4)	171		
- Ubl4A	177		
- Bag6 (Bat3)	133		
SGTA	549	C	
TRC40 (Asna1, Get3)	381	C	
*TA receptor*		ERM	
- CAML (CAMLG, Get2)	5		
- WRB (CHD5, Get1)	4		Congenital Heart Disease
*ERM protein complex*		ERM	
- EMC1	124		
- EMC2	300		
- EMC3	270		
- EMC4	70		
- EMC5 (MMGT1)	35		
- EMC6 (TMEM93)	5		
- EMC7	247		
- EMC8	209		
- EMC9	1		
- EMC10	3		
*#TMCO1 complex*		ERM	
- TMCO1 ##	2013		Glaucoma, Cerebrofaciothoracic Dysplasia
- Nicalin	99		
- TMEM147	21		
- CCDC47 (Calumin)	193		
- NOMO	267		
*PAT complex*		ERM	
- PAT10 (Asterix)			
- CCDC47 (Calumin)	193		
PEX19	80	C	Zellweger Syndrome
PEX3	103	ERM	Zellweger Syndrome
*#Sec61 complex ##*		ERM	
- Sec61α1	139		Diabetes **, CVID, TKD, Neutropenia
- Sec61β	456		PLD, Colorectal Cancer
- Sec61γ	400		GBM, Hepatocellular Carcinoma, RCC
#Sec62 (TLOC1)	26	ERM	Breast-, Prostate-, Cervix-, Lung-Cancer et al.
*ER Chaperones*			
Sec63 (ERj2)	168	ERM	PLD, Colorectal Cancer et al.
#ERj1 (DNAJC1)	8	ERM	
ERj3 (DNAJB11)	1001	ERL	Polycystic Kidney Disease (PKD)
ERj4 (DNAJB9)	12	ERL	
ERj5 (DNAJC10)	43	ERL	
ERj6 (DNAJC3, p58^IPK^)	237	ERL	Diabetes, Neurodegeneration
ERj7 (DNAJC25)	10	ERM	Hyperinsulinismus, Allergic Asthma
ERj8 (DNAJC16)	24	ERM	
ERj9 (DNAJC22)		ERM	
BiP (Grp78, HSPA5)	8253	ERL	Hemolytic Uremic Syndrome (HUS)
Grp170 (HYOU1)	923	ERL	Immunodeficincy & Hypoglycemia
Sil1 (BAP)	149	ERL	Marinesco-Sjögren-Syndrome (MSS)
Grp94 (CaBP4, Hsp90B1)	4141	ERL	
PPIB (Cyclophilin B)	1289	ERL	
FKBP2 (FKBP13)	894	ERL	
PDIA1 (PDI, ERp59)	3624	ERL	Cole-Carpenter Syndrome
PDIA2 (PDIp)		ERL	
PDIA3 (ERp61, Grp57)	3730	ERL	
PDIA4 (ERp72, CaBP2)	2173	ERL	
PDIA5 (PDIR)	37	ERL	
PDIA6 (P5, CaBP1)	3001	ERL	
PDIA9 (ERp29)		ERL	
Calreticulin (CaBP3, ERp60)	14521	ERL	
#Calnexin_palmitoylated_	7278	ERM	
#TRAM1	26	ERM	
TRAM2	40	ERM	
*#TRAP complex*		ERM	
- TRAPα (SSR1)	568		
- TRAPβ (SSR2)			
- TRAPγ (SSR3)	1701		CDG, Hepatocellular Carcinoma
- TRAPδ (SSR4)	3212		CDG
#RAMP4 (SERP1)		ERM	
*#Oligosaccharyltransferase (OST-A)*		ERM	
- RibophorinI (Rpn1)	1956		
- RibophorinII (Rpn2)	527		
- OST48	273		CDG
- Dad1	464		
- OST4			
- TMEM258			
- Stt3A *	430		CDG
- DC2			
- Kcp2			
*Oligosaccharyltransferase (OST-B)*		ERM	
- RibophorinI (Rpn1)	1956		
- RibophorinII (Rpn2)	527		
- OST48	273		CDG
- Dad1	464		
- OST4			
- TMEM258			
- Stt3B*	150		CDG
- TUSC3			CDG
- MagT1	33		
*Signal peptidase (SPC-A)*		ERM	
- SPC12	2733		
- SPC18 * (SEC11A)			
- SPC22/23	334		
- SPC25	94		
*Signal peptidase (SPC-C)*		ERM	
- SPC12	2733		
- SPC21 * (SEC11C)			
- SPC22/23	334		
- SPC25	94		
*GPI transamidase (GPI-T)*		ERM	
- GPAA1	9		
- PIG-K	38		
- PIG-S	86		
- PIG-T	20		
- PIG-U	42		
*Additional modifying enzymes*			
ALG8	10	ERM	CDG, PLD
UGGT	232	ERL	
Glucosidase IIα (GIIα)		ERL	PLD
Glucosidase IIβ (PRKCSH, GIIβ)		ERL	PLD
Proly-4-hydroxylase α (4-PH)		ERL	
Proly-4-hydroxylase β (PDI)	3624	ERL	Cole-Carpenter Syndrome
SUMF1	23	ERL	Multiple Sulfatase Deficiency
SUMF2	386	ERL	Multiple Sulfatase Deficiency
#p34 (LRC59, LRRC59)	2480	ERM	
#p180 (RRBP1)	135	ERM	Hepatocellular Carcinoma, Colorectal Cancer
Kinectin 1 (KTN1)	263	ERM	

Protein classes or complexes are characterized by italics, subunits of complexes are identified by hyphens, alternative names of components or subunits are given in parentheses. Abundance is given in nM according to Hein et al. (see Data Availability) C, cytosol; CDG, Congenital disorder of glycosylation; CVID, Common variable immunodeficiency; ERL, ER lumen; ERM, ER membrane; GBM, Glioblastoma multiforme; PLD, polycystic liver disease; RCC, renal cell carcinoma; SUMF, sulfatase modifying factor or formylglycine generating enzyme; TKD, Tubulo-interstitial kidney disease; UGGT, UDP-glucose-glycoprotein glucosyltransferease. *, catalytically active; **, in mice; #, ribosome associated; ##, ion channel activity. We note that (i) Calnexin, ERj1, ERp72, P5, Sec61β, Sec63, SRα, TRAM1, and TRAPα were shown to be subject to phosphorylation, (ii) BiP, Calnexin, Calreticulin, CCDC47, Grp94, Sec62 and TRAPα are calcium binding proteins, (iii) Sec63, TRAPα, and TRAPβ were predicted to comprise immunoglobulin-like β sandwich domains in the cytosol and ER lumen, respectively, (iv) hSND3, WRB, EMC3, TMCO1, Sec62, Sec63, and ERj1 were predicted to comprise cytosolic coiled coil domains, and (v) WRB, EMC3, and TMCO1 are OXA1-homologs.

**Table 2 cells-10-01036-t002:** Sec61 channel inhibitors.

Inhibitor	Binding Site in Sec61α	Effect	Linked Disease(s)
Apratoxin A	T86 (TMH2), Y131 (TMH5)	non-selective	
CAM741		selective inhibition	
Coibamide A		non-selective	
Cotransin CT8	R66 (loop 1), G80 (TMH2), S82 (loop 1), M136 (TMH3)	selective inhibition	
Eeyarestatin ES24		non-selective Ca^2+^ leak inducing	
Exotoxin A	N-terminus	non-selective Ca^2+^ leak blocking	Pneumonia, Sepsis
Ipomoeassin F		non-selective	
Mycolactone	R66 (loop 1), S82 (loop 1)	selective inhibition Ca^2+^ leak inducing	Buruli ulcer

**Table 3 cells-10-01036-t003:** Human hereditary Sec61 channelopathies and related diseases.

Disease	Linked Gene(s)	Sec61 Effect	MIM
ADPKD	*DNAJB11*	Ca^2+^ leak induction *	
	*PKD1*, *PKD2*, *UMOD*		
ADPLD	*ALG8*, *GANAB*, *PRKCSH*,		
	*SEC61B*, *SEC63*	selective transport inhibition	617004
ADSCN	*ELANE*, *JAGN1*, *SEC61A1*	transport inhibition	
		Ca^2+^ leak induction	
ADTKD	*HNF1B*, *MUC1*, *REN*,	selective transport inhibition	
	*SEC61A1*, *UMOD*	Ca^2+^ leak induction	617056
CDG	*ALG8*, *SSR3*, *SSR4*, *OST48*,	selective transport inhibition	
	*STT3A*, *STT3B*, *TUSC3*		
CVID	*CD19, CD20, CD21*,	transport inhibition	
	*CD80*, *SEC61A1*	Ca^2+^ leak induction	
Diabetes	*DNAJC3*, *INS*	Ca^2+^ leak induction *	
MSS	*SIL1*		248800

The proteins, which are coded by the underlined genes, are discussed in detail in this review. ADPKD, autosomal dominant polycystic kidney disease; ADPLD, autosomal dominant polycystic liver disease; ADSCN, autosomal dominant severe congenital neutropenia; ADTKD, autosomal dominant tubulo-interstitial kidney disease; CDG, congenital disorder of glycosylation; CVID, common variable immunodeficiency; MSS, Marinesco–Sjögren-Syndrome. *, expected.

## Data Availability

Protein abundances in HeLa cells, given in Table 1, were reported by Hein et al. [342]. SP sampling in the cytosolic funnel of the Sec61 channel was brilliantly simulated and visualized by Zhang and Miller [136,343]. The conformational changes of the modelled Sec61 complex, shown in Figure 2, were previously morphed [344]. The Ca^2+^-permeability of the open Sec61 channel as observed by live cell Ca^2+^ imaging can be seen in the video file of [345]. cBio Portal for cancer genomics is found in [346].

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
