# Peer review of "Complexity and Specificity of Sec61-Channelopathies: Human Diseases Affecting Gating of the Sec61 Complex"

_cells, 2021, doi:10.3390/cells10051036_

Round 1
Reviewer 1 Report
The Sec61 translocon forms the entry gate into the ER that is used by hundreds of secreted proteins. This review provides a very comprehensive and detailed overview about the molecular structure of the translocon and its intricate interplay with the luminal chaperone system. Moreover, this manuscript describes the relevance of the translocon for diseases in great depth. It is written by experts in the field and will be a very important source of information for many readers. I really enjoyed reading it and the strong focus on disease-related aspects sets it apart from other reviews in the literature. However, a few minor points should be considered.
- Some of the figures are of low quality. E.g. Fig. 3a shows nice structures of the Sec61 complex but the blobs and squares for BIP is little informative. It is unclear whether the BiP cycle shown has anything to do with the two states of the Sec61 channel. Does Sec62/63 and TRAP leave the complex upon translocation as shown here? Is calcium release really the trigger for channel opening as shown? The authors should carefully revise the figures as they carry important information for many readers.
- Figures cont.: Many details of the figures are impossible to see due to question marks and weird signs which presumably were caused by pdf conversion. This needs to be fixed.
- Figures cont.: Figure 2 also needs more thought. Why are some proteins green or yellow. How does the text underneath the membrane relate to the shown signals? Authors show nine examples and list seven categories: It is unclear which of these corresponds to which group. This figure needs to be improved.
- Figures cont.: Figure 5 is very nice, but the many abbreviations make it very difficult to extract the information. Most readers will not be familiar with MSS, GBM, ADSCN etc. I found the glossary in the end, but it would be better to either spell out the names in the figure or in the legend.
- Why are some protein names underlined in table 2?
- Why do authors show the picture of a patient in Fig. 7? It is not clear to me how this informative. The respective sentence does also not give any reference to this picture: ‘Interestingly already in 1993, an enrichment of BiP within the nuclear envelope was demonstrated in the musculature of rabbits and this sub-cellular localization within muscle fibers could even be recapitulated in humans (Figure 7)’
- For the introduction part it would be good to provide some overview about the EMC and GET (CAML/WRB) complex as they will be of interest for many readers. They are mentioned throughout the text, but some more structured information at the beginning of the article would recommended.
Author Response
We are grateful to the two reviewers for their help in improving our manuscript.
They did a fantastic job and we are convinced that our following their advice indeed improved the manuscript a lot.
We have replaced all figures by high resolution tiff files and are submitting the marked-up manuscript and a pdf file of the revised manuscript in order to make the revisions visible and to avoid conversion problems by MDPI, respectively.
Response to reviewer 1:
- We removed old Figure 3a and replaced it by new Figure 2, which is used to introduce the concept of Sec61 channel gating, as suggested by reviewer 2. In addition, we extended the legends to several figures to eliminate apparent mis-interpretation that was caused by our poor text.
- We were not aware of theses problems, which may have occurred during pdf creation by MDPI. We apologize.
- We improved the old figure 3 (now 4) and its legend to prevent mis-interpretation that was caused by our poor text.
- We extended the legend to old Figure 5 (now 6), as suggested.
- We explain this in the new legend to the table.
- As we have indicated by changes in text and legend, old Figure 7 (now 8) shows hallmarks of MSS patients.
- We have added a section on membrane protein insertases on new page 16 (lines 471 through 496), as suggested.
Reviewer 2 Report
This is a comprehensive and updated review on the function and regulation of the Sec61p translocation complex. The reviewers included a massive amount of information on the basic biology and biochemistry concerning the structure and function of Sec61p during both protein translocation and Ca2+homeostasis, as well as its roles in physiology and pathology. The content provides a timely update for some of the most recent findings in this field, while providing a reasonable summary of information on the structure and function of Sec61p from previous decades of research in both protein translocation and Ca2+homeostasis. The role of Sec61p in Ca2+storage in the ER and in human diseases are not often discussed in other reviews on Sec61p. As such, the review makes a welcome and valuable contribution.
There are some conceptual points of confusion in this review that need to be clarified. The presentation could also be significantly improved by a better organization of the article as well as improvements in the writing. These points are detailed below.
- A recurring focus of the review is the concept of Sec61p ‘gating’. However, this term was never clearly defined. I infer, rather from my background than from the review, that the term refers to the conformational change of Sec61p from the closed to the open state, which includes opening of the lateral gate at TM2/7 and removal of the plug in the Sec61p pore.
It was also unclear what experimental evidence were used to infer ‘gating’ in various sections throughout the article. In the cases of SP and Sec62/63p, it was relatively clear that structural information were used. However in the cases pertaining to TRAP, TRAM, and BiP (section 2.3), the evidence seems to be mostly based on stimulation of translocation, which could be interpreted by alternative models. In these cases, gating seems to be used interchangeably with translocation, but surely a defect in translocation does not necessarily imply a ‘gating’ defect. For example, BiP could assist in translocation by helping to ‘pull’ the precursor protein into the ER. For another example, Sec62/63 and TRAP/TRAM could displace one another to give co- or post-translational substrates better access to Sec61p.
The issue becomes more confusing in the discussions on Ca2+homeostasis, as the ‘open’ state for protein translocation may not be the same as the ones that mediate Ca2+permeability. Indeed, the fact that BiP activates protein translocation but inhibits Ca conductance implies that either the Sec61p complex or the BiP interaction with Sec61p is different in the two activities. In addition, alternative models, such as steric block of the channel upon binding of BiP or CaM, could easily explain the channel recording data and blocking of Ca2+permeability.
In summary, both the concepts of ‘gating’ and experiments used to probe it need to be clarified, and nuances in this phenomenon need be addressed. This may be best done as a summary paragraph at the beginning of the review, and the nuances could be further discussed as individual situations are described.
- The review contains an extensive amount of information, but the content is meandering and difficult to follow during the reading. Part of the problem is that some of the information is tangentially related to the main thrust of the article. I suggest that these material could be placed as tables, footnotes, or side notes to allow a clearer conceptual development in the main text. Specific content that falls into this category are listed below:
- Lines 49-62: summarize as a table.
- Lines 105-111: can be replaced by a short summary, such as “as reviewed in other articles in this issue, BiP is also a key player in signaling pathways that report on ER protein and energy homeostasis”.
- Lines 589-615 (ends with ‘see below’): This paragraph can be removed or converted to a footnote.
- Lines 714-724: can be converted to a footnote.
- Section 5.3: this is a long section describing many BiP- and cochaperone-related diseases.However, BiP has numerous roles other than assisting the function of Sec61p, most notably as the major ER-resident chaperone, and most of the information in this section appears to be unrelated to Sec61p functions. Since the article and section 5 is on ‘translocation’, some distillation of information is needed here.
- Most the paragraphs are overly long. Some are writing issues mentioned in (4), whereas in others, the content can be separated to improve clarity and conceptual progression in the writing. In addition, some paragraphs are more appropriate in a different section of the article. Specific suggestions are listed below:
- Lines 204-224 (ends with ‘import’): This can be another paragraph.
- Lines 224-232 (starts with “Notably,”): Start another paragraph here. This new paragraph can be merged with the next paragraph (lines 233-241).
- Lines 163-241: This is a very long introduction of section 2. Need at least one subtitle summarizing these contents.
- Lines 361-379: This paragraph seems better belong to the next sub-section (2.3).
- Section 2.3: a more appropriate subtitle would be “Auxiliary factors that regulate Sec61p”.
- In general, the writing could significantly benefit from input and revisions from a professional science writer. This includes both the content organization within a paragraph, unnecessary repetitions of content within a paragraph and in different parts of the article, and grammar and syntax issues. An incomplete list of these issues are in point (5) below. Nevertheless, it will be worthwhile to have a professional writer address them in a more complete, systematic way.
- Some of the writing issues are listed below:
- At least in the PDF version I can see, almost all figures have strange symbols, in some cases to the point of obfuscating the legend. ‘plus’ is overused in the article and can be replaced by other conjunctions.
- Lines 96-104: can be removed, as the same content was repeated later in the article.
- The first reference to Figure 4 is in lines 85-88 and should be cited there.
- Figure 1: there are panel labels in the legend but not the figure.
- Line 1088: remove ‘and NEFs’ (they are Hsp70 cochaperones).
Author Response
We are grateful to the two reviewers for their help in improving our manuscript.
They did a fantastic job and we are convinced that our following their advice indeed improved the manuscript a lot.
We have replaced all figures by high resolution tiff files and are submitting the marked-up manuscript and a pdf file of the revised manuscript in order to make the revisions visible and to avoid conversion problems by MDPI, respectively.
Response to reviewer 2:
- We have replaced old Figure 3a by new Figure 2 to introduce the concept of Sec61 channel gating early in the text, as suggested, and extended the figure legend to explain how channel opening and closing are assayed.
- We have changed the text to make it easier to digest but refrained from adding an additional table in order to avoid distraction from the main thrust.
We have changed the text as suggested.
We have considerably shortened the text on ERjs (page 19).
We have considerably shortened the text on cross-presentation (page 20).
- We have followed your advice for improvements, as suggested. Thank you very much for your help.
- We involved the wive of one author who is a native English speaker in improving the text. Furthermore, we shortened the text on more pages than was requested. However, we kept some oft he repetitions in the bleieve that this chapter may also be read by MD, who are not familiar the topic and, therefore, need to be introduced by stepwise increase in details.
- We were not aware of theses problems, which may have occurred during pdf creation by MDPI. We apologize.
We have eliminated many „plus“.
We removed the text, as suggested.
Old Figure 4 was turned into new Figure 3 and cited, as suggested.
The panel labels were moved to make them visible.
In our opinion, it is a matter of debate if a NEF is a co-chaperone. We prefer to differentiate between co-chaperones and NEFs and, therefore, referained from removing „and NEFs“.